# A Comprehensive Investigation of Macro-Composition and Volatile Compounds in Spring-Picked and Autumn-Picked White Tea

**DOI:** 10.3390/foods11223628

**Published:** 2022-11-14

**Authors:** Cheng Zhang, Chengzhe Zhou, Kai Xu, Caiyun Tian, Mengcong Zhang, Li Lu, Chen Zhu, Zhongxiong Lai, Yuqiong Guo

**Affiliations:** 1College of Horticulture, Fujian Agriculture and Forestry University, Fuzhou 350002, China; 2Tea Industry Research Institute, Fujian Agriculture and Forestry University, Fuzhou 350002, China; 3Institute of Horticultural Biotechnology, Fujian Agriculture and Forestry University, Fuzhou 350002, China

**Keywords:** white tea, season, volatile compounds, widely targeted volatilomics method

## Abstract

The flavour of white tea can be influenced by the season in which the fresh leaves are picked. In this study, the sensory evaluation results indicated that spring-picked white tea (SPWT) was stronger than autumn-picked white tea (APWT) in terms of the taste of umami, smoothness, astringency, and thickness as well as the aromas of flower and fresh. To explore key factors of sensory differences, a combination of biochemical composition determination, widely targeted volatilomics (WTV) analysis, multivariate statistical analysis, and odour activity value (OAV) analysis was employed. The phytochemical analysis showed that the free amino acid, tea polyphenol, and caffeine contents of SPWTs were significantly higher than those of APWTs, which may explain the higher umami, smoothness, thickness, and astringency scores of SPWTs than those of APWTs. The sabinene, (2E, 4E)-2, 4-octadienal, (-)-cis-rose oxide, caramel furanone, trans-rose oxide, and rose oxide contents were significantly higher in SPWTs than in APWTs, which may result in stronger flowery, fresh, and sweet aromas in SPWTs than in APWTs. Among these, (2E,4E)-2,4-octadienal and (-)-cis-rose oxide can be identified as key volatiles. This study provides an objective and accurate basis for classifying SPWTs and APWTs at the metabolite level.

## 1. Introduction

Tea is the most popular non-alcoholic beverage in the world due to its pleasant taste, attractive aroma, and beneficial health properties [1,2,3,4,5,6]. The quality of tea is determined by various distinctive secondary metabolites, belonging mainly to polyphenols, flavonoids, free amino acids, and volatile olfactive compounds (VOCs) [7]. Moreover, tea polyphenols enhance the bitterness of tea infusions, flavonoids supply the bitter taste, free amino acids are the source of the umami taste, and VOCs give the tea a wide variety of aroma qualities [8,9,10]. The biosynthesis of secondary metabolites is affected by many factors such as tea species, climate environment, soil, manufacturing processes, and picking season. Among them, the picking season is one of the key factors affecting the content of secondary metabolites. For example, the content of theanine, kaempferol-glycoside epicatechin, and catechin in fresh leaves in spring-picked (SP) is higher than that in autumn-picked (AP) [11,12,13]. Black tea in SP has higher caffeine and catechin contents than in AP [14]. The contents of (-)-epigallocatechin gallate (EGCG) and (-)-epicatechin gallate (ECG) in oolong tea in SP were lower than those in AP, while the differences in catechin gallate (CG) and total catechins (TC) contents were not obvious [15,16]. The content of free amino acids in SP green tea is higher than that in AP green tea, while the content of flavonoids is lower than that of AP green tea [13,17]. The different contents of these compounds in SP and AP teas may have contributed to the different tastes. Fifteen VOCs, including trans-β-ionone, nonanal, and dimethyl sulfide, were found to be the main VOCs that distinguished SP Xinyang Maojian green teas from AP teas [18]. Linalool, β-damascenone, and benzeneacetaldehyde were considered to be the main VOCs that distinguish Yingde black teas [19]. A total of 12 aroma substances, including cis-jasmone, benzyl alcohol, and (E)-2-octenal, are key differentiating compounds between green teas of different seasons [20]. Linalool has a “floral” odour; nonanal and (E)-2-octenal imparted a “green, fatty, or tallow” note; cis-jasmone was considered to have a “characteristic floral jasmine” odour; and benzyl alcohol has a “burning” taste and a “faint aromatic” odour [21,22,23,24]. The different content of these VOCs in tea picked in different seasons contributes to the distinctive aroma of tea. Therefore, the picking season is an important factor in the formation of the flavour quality of tea, which has a significant impact on the taste and aroma of tea.

White tea is one of the six major types of tea in China; it originated in Fujian province and is made from tea leaves that have been subjected to a long withering and drying process [24]. White tea is characterized by umami and a slightly sweet taste along with a fresh and green odour [25]. It has been widely recognized for its health-promoting properties, including antioxidant, hypoglycemic, and lipid-lowering [3,5,26,27]. Its potential health benefits and unique flavour have led to a growing market share and widespread scientific interest [28]. Depending on the selection criteria and raw material used to make the tea, white tea can be divided into four types: silver needle white, white peony, Gong Mei, and Shou Mei teas [29]. Silver needle white tea is produced only in early spring, whereas white peony and Shou Mei teas are produced in both late spring and early autumn. The main difference between white peony and Shou Mei teas is the maturity of the harvested leaves. White peony tea is made with one bud with the first to second tender leaves, whereas Shou Mei is made with one bud with the third to fifth leaves [30]. There are notable differences in the aroma and taste between spring-picked white tea (SPWT) and autumn-picked white tea (APWT). The taste of SPWTs is more umami, and the aroma in the review glass is more lasting at temperatures of 75 °C, 45 °C, and room temperature [31]. However, most studies on white tea have focused on the chemical qualities of different types, grades, and storage ages, whereas few studies have been conducted on the relationship among macroscopic components, VOCs, and picking seasons [28,32,33,34,35]. The aroma and taste of tea are determined largely by the amount and composition of different chemicals [25]. With a deeper understanding of the reasons for the differences between SPWTs and APWTs, production can be better guided. Therefore, it is necessary to comprehensively investigate the differences in macro-composition and VOCs between SPWTs and APWTs, and systematically study the relationship among taste quality, macro-composition, and VOCs.

To investigate the differences between SPWTs and APWTs, three consecutive years (2019–2021) of SPWTs and APWTs were selected as experimental samples. Annual differences in raw material quality and differences in chemical transformation and storage time contribute to these changes. To minimise their impact on macro-composition and VOCs, we used fresh leaves from the same tea garden as raw materials, with the same variety and the same picking standard, and produced the same grade of white peony and Shou Mei teas under the same conditions. In addition, in the data analysis, the same type of white tea in SP and AP tea in the same year was selected for analysis. The widely targeted volatilomics (WTV) method was used to identify the volatile components of SPWTs and APWTs. Compared with the traditional aroma analysis method, WTV has the advantages of high sensitivity, high annotation coverage, and good reproducibility [36]. Macro-composition quantification, multivariate statistical analysis, and odour activity value (OAV) analysis were also performed. This study will provide an objective and accurate theoretical basis for differentiating SPWTs and APWTs. 

## 2. Materials and Methods

### 2.1. Chemicals

Folinphenol, sodium carbonate, hydrochloric acid, sulfuric acid, disodium hydrogen phosphate, potassium dihydrogen phosphate, ninhydrin, aluminium chloride, sodium bicarbonate, oxalic acid, ethyl acetate, and n-butanol were purchased from Sinopharm Chemical Reagent Co. Ltd. (Shanghai, China). The plant-soluble sugar content test kit was purchased from Suzhou Keming Biotechnology Co. Ltd., Suzhou, China. Sodium chloride, n-hexane (purity > 95%), and 3-hexanone-2,2,4,4-d4 (purity > 95%) were purchased from Sinopharm, Merck, and Sigma-Aldrich, Missouri, USA, respectively.

### 2.2. Materials

Samples of 12 white teas were made from fresh leaves of clonally propagated *Camellia sinensis* ‘Fudingdahaocha’ grown in the same tea field of Fuding Minghai Tea Co., Ltd., in Fuding County, China (119°59′ E, 27°19′ N). White peony tea was made with one bud and the first to the second tender leaves, whereas Shou Mei tea was made with one bud with three to five leaves and then processed using the same procedures according to the National Standard of China (GB/T 32743-2016). In brief, fresh tea leaves were withered at 30 ± 5 °C and 50 ± 5% relative humidity for 48 h, then dried at 80 °C for 60 min. In the spring and autumn of 2019–2021, SPWTs and APWTs samples were prepared. Spring-picked was referred to as SP, and autumn-picked was called AP. White teas picked in spring were called SPWTs, and those picked in autumn were called APWTs. White peony teas and Shou Mei picked in spring were termed CMD19-21 and CSM19-21, respectively, while those picked in autumn were correspondingly named QMD19-21 and QSM19-21. The number following CMD, QMD, CSM, and QSM represented the production years, corresponding to 2019, 2020, and 2021, respectively. The number after the year indicates a biological duplicate (Table 1). Tea samples were stored in the same warehouse in a controlled environment (20 ± 5 °C and 50 ± 5% relative humidity). Each sampling was performed in triplicate.

### 2.3. Sensory Evaluation

Tea samples were evaluated and scored by a panel of 12 expert panelists (six women and six men), who were professional assessors and trained according to the National Professional Standards for Tea Sensory Evaluation (Profession Code: 6–02-06–11, China) and had more than 10 years of descriptive sensory analysis experience with tea. Tea infusions were prepared according to national standards (GB/T 23776-2018): each tea sample (3 g) was brewed with 150 mL boiling water for 5 min and discarded, after which tea infusions (30 mL) labelled with a three-digit code were presented to each panelist in a randomised order (Figure 1). Then, panelists provided a taste description (smoothness, sweetness, thickness, astringency, and umami) and aroma description (flower, sweet, and fresh) for each tea sample and rated them on a scale from 0 (not present) to 9 (very strong). “Smoothness” is a pleasant taste with a silky softness and a buttery smoothness. “Sweetness” is the taste connected with sugary foods. “Thickness” was defined as the pleasure of having a coating on the tongue. “Astringency” is the sensation of dryness and contraction of the tongue and soft palate. “Umami” is a delicious flavour induced by several amino acids. “Flower” is defined as a blend of natural floral odour. “Sweet” is the smell of sucrose solution. “Fresh” is a fresh hay-like aroma [30,35].

### 2.4. Macro-Composition Quantification

#### 2.4.1. Quantification of Water-Extractable Substances

Water-extractable substances were quantified using a different method based on the National Standard of China (GB/T 8305-2013); a tea infusion was made with 450 mL of boiling water and 3 g of tea. The amount of 50 mL of tea infusion was pipetted into an evaporating dish, heat-dried and evaporated, dried in an oven (120 °C, 1 h), cooled to room temperature, and weighed for calculation. 

#### 2.4.2. Quantification of Polyphenols

Polyphenols were quantified using the colourimetric method with Folin-Ciocalteu reagent based on the National Standards of China (GB/T 8312-2018). Specifically, 0.2 g of tea was added to 4.5 mL of 70% methanol (preheated to 70 °C); after stirring and cooling, it was centrifuged at 3500 r/min for 10 min, and the supernatant transferred to a 10 mL test tube. The amount of 1 mL of the supernatant was taken into a volumetric flask, and 5 mL of Foline-Ciocalteau reagent and 4 mL of 7.5% Na_2_CO_3_ were added, and it was shaken for 8 min and made volume with pure water. The absorbance of the mixture was measured at 765 nm.

#### 2.4.3. Quantification of Caffeine

Caffeine was quantified using the UV spectrophotometric method based on the National Standard of China (GB/T 8312-2013). Firstly, a 2 mL tea infusion was mixed with 4 mL of 0.01 mol/L hydrochloric acid and 1 mL basic lead acetate solution in a 100 mL volumetric flask. After adding up to volume with water and being mixed, the solution was filtered. The 12.5 mL filtrate was taken into a 25 mL volumetric flask and added with 0.05 mL of 4.5 mol/L sulfuric acid to eliminate residual lead ions in the solution. After adding up to the volume with water and shaken, the mixed solution was filtered. The absorbance of the second filtrate was measured spectrophotometrically at 274 nm. 

#### 2.4.4. Quantification of Total Free Amino Acids

The total free amino acids were quantified using the colourimetric method based on the National Standard of China (GB/T 8314-2013), with minor modifications [37]. A 1 mL tea infusion and 0.5 mL phosphate buffered saline were put into a 25 mL test tubes, then 0.5 mL ninhydrin was also added. The tubes were placed in a boiling water bath for 10 min and then immediately cooled to room temperature in an ice bath. Finally, the absorbances of the samples were measured spectrophotometrically at 570 nm. Flavonoids were quantified using the colourimetric method with aluminium trichloride [38]. 

#### 2.4.5. Quantification of Soluble Sugars

Soluble sugars were quantified using the plant-soluble sugar content test kit (Grace Biotechnology Co., Ltd., Suzhou, China) according to the manufacturer’s instructions. The content was calculated according to the following formula:$$\mathrm{Soluble}\;\mathrm{sugar}\;(\mathrm{mg}\;g^{-1})=2.34\;\text{×}\;(\Delta A+0.07)/W$$

#### 2.4.6. Quantification of Total Content of Theabrownins (TBs), Theaflavins (TFs), and Thearubigins (TRs)

The total content of TBs, TFs, and TRs was determined using a previously discussed method [38,39]. Briefly, a total of 3 g of dried tea was extracted with 125 mL boiling distilled water in a boiling water bath for 10 min and filtered, followed by cooling to room temperature. The amount of 50 mL of tea filtrate was pipetted in 50 mL of ethyl acetate and shaken for 5 min, and the layers were separated after equilibration. A 0.8 mL portion of the ethyl acetate layer was made to a volume of 10 mL with 95% ethanol (solution A). Another 15 mL of the ethyl acetate layer was shaken with 15 mL of NaHCO_3_ solution (2.5%) for 30 s, standing until the layers separated. The amount 1.6 mL of ethyl acetate layer was pipetted, and the volume fixed to 10 mL with 95% ethanol (solution C). A 0.8 mL sample of portions of the aqueous layer was diluted to 10 mL with 2.4 mL distilled water, 0.8 mL saturated oxalic acid solution (10.2%, m/v), and 6 mL of 95% ethanol (solution D). A 25 mL amount of tea filtrate was pipetted and mixed with 25 mL of butyl alcohol. After shaking for 3 min, the layers were separated after equilibration. A 0.4 mL sample of the aqueous layer (second) was made to volume of 10 mL with 0.4 mL of the saturated oxalic acid solution, 1.2 mL of distilled water, and 95% ethanol (solution B). The absorbance of solutions A, B, C, and D at 380 nm was measured spectrophotometrically using 95% ethanol as a blank. The results were calculated using the following formula:$$\mathrm{TFs}=\;\;\frac{E_{c}\times 2.25}{dried\;weight\left(\%\right)}\times 100\%$$



$$\text{TRs}=\frac{7.06\times \left(2E_{A}+2E_{D}-E_{C}-2E_{B}\right)}{dried\;weight\left(\%\right)}\times 100\%$$


$$\mathrm{TBs}=\frac{2E_{B}\times 7.06}{dried\;weight\left(\%\right)}\times 100\%$$



### 2.5. Identification and Analysis of Volatiles

Volatiles were extracted from the samples using the headspace solid-phase microextraction (HS-SPME) method with three replicates for each assay. After sampling, volatile analysis was performed using an Agilent Model 8890 GC equipped with a 30 m × 0.25 mm × 0.25 μm DB-5MS (5% phenyl-polymethylsiloxane) capillary column and a 7000D mass spectrometer (Agilent, Santa Clara, CA, USA). HS-SPME and gas chromatography–mass spectrometry (GC-MS) analysis procedures were performed, as previously described [40], with minor modifications. The temperature was programmed at 40 °C for 3.5 min, increased at 10 °C/min to 100 °C, then increased at 7 °C/min to 180 °C, and finally increased at 25 °C/min to 280 °C and was held for 5 min. The flow rate of the carrier gas (helium, 99.999%) was 1.2 mL/min. The MS spectrometer was operated in electron impact mode with electron an energy of 70 eV and a scan range of *m*/*z* 50–500. The ion source and mass spectrum transferline temperatures were 230 °C and 280 °C, respectively. The specific GC-MS acquisition conditions are listed in Appendix A. During instrumental analysis, a quality control sample (prepared from the sample mix) was inserted into every tenth assay samples to monitor the repeatability of the analysis process. VOCs were identified using the WTV method. The volatile compounds were identified by comparing the mass spectra with the data system library (MWGC) and linear retention index. Briefly, one quantitative ion and two to three qualitative ions were selected for each compound. All ions to be detected in each group were detected separately in the order of peak appearance, if the retention times of the detected peaks were consistent with the standard reference and if all the selected ions appeared in the mass spectra of the samples after subtraction of background, the substance was determined [36]. Chemical structures, names, and aromas of VOCs were obtained from PubChem (https://pubchem.ncbi.nlm.nih.gov, accessed on 10 April 2022) and the Good Scents Company Information System (http://www.thegoodscentscompany.com accessed on 10 April 2022). The internal standard used was 3-hexanone-2,2,4,4-d4 (10 uL, 50 ug/mL), and the relative content of each volatile compound was calculated using the following formula:(1)$$C_{i}=\frac{\frac{A_{i}}{A_{is}\times m_{is}}}{m_{i}}$$
where $C_{i}$ is the mass concentration of each component (μg kg^−1^), $m_{is}$ is the mass of the internal standard (μg), $A_{i}$, and $A_{is}$ are the chromato-graphic peak area of each component and internal standard, respectively, and $m_{i}$ is the mass of the sample powder (kg).

The *OAV* of the odour-active compounds was calculated as the ratio of the concentrations and thresholds in the water of these compounds. Compounds with an *OAV* > 1 are considered to contribute to the aroma of white tea [41]. The formula for calculating the *OAV* value is as follows:(2)$$OAV_{i}\;=\frac{C_{i}}{OT_{i}}$$
where $C_{i}$ (µg kg^−1^) is the VOC content, and $OT_{i}$ (µg kg^−1^) is the aroma threshold of the volatile components in water [42].

### 2.6. Statistics

All analyses were repeated three times. Unsupervised principal component analysis (PCA) was performed using the statistical function prcomp in the R package (version 3.5.1, http://www.r-project.org/, accessed on 6 November 2022). A radar plot was drawn using Origin (v2021, OriginLab Corporation, Northampton, MA, USA). Orthogonal partial least squares–discriminant analysis (OPLS-DA) was performed using SIMCA software (version 13.0; Umetrics, Umea, Sweden). Statistical significance was determined using one-way analysis of variance (ANOVA) and Duncan’s multiple range test using SPSS (version 25.0, IBM, Armonk, NY, USA). 

## 3. Results and Discussion

### 3.1. The Taste and Aroma of SPWTs Are Stronger Than Those of APWTs

To explore the taste (including smoothness, sweetness, thickness, astringency, and umami) and aroma (including flowery, sweet, and fresh) characteristics of SPWTs and APWTs, quantitative descriptive analysis, a useful method, was performed (Figure 2) [43,44]. In the comparison of the white peony tea groups, higher taste and aroma scores were found in the SP samples than in the AP samples (Figure 2A,C). However, taste characteristic scores differed in the Shou Mei tea group, with no similar pattern emerging for sweetness scores, whereas other scores were higher in SP samples than in AP samples (Figure 2B,D). As the storage time increased, the freshness and astringency gradually decreased, whereas the intensities of smoothness and thickness increased, which is consistent with the findings of previous studies [35]. The significant difference analysis of the taste and aroma attribute scores shows that when comparing the SPWTs and APWTs groups of the same vintage, SPWTs were stronger than APWTs in terms of the taste of umami, smoothness, astringency, and thickness as well as the aromas of flower and freshness (Appendix A). These results indicate that the sensory characteristics of SPWTs and APWTs differ. 

### 3.2. Analysis of Macro-Composition of SPWTs and APWTs

The relative total tea polyphenols, caffeine, free amino acid, and flavonoids contents in SPWTs are higher than those in APWTs. The relative contents of water-extractable substances and soluble sugars were not significantly different in SPWTs and APWTs (Figure 3).

#### 3.2.1. Water-Extractable Substances

Water-extractable substances are soluble substances extracted from tea leaves using boiling water under defined conditions [45]. The main substances in the water-extractable, including polyphenols such as caffeine, catechins, and free amino acids determine the consistency and taste of the tea infusion and are considered to be one of the most important indicators of tea quality [46]. We found that in the white peony tea group, there were no significant differences in the content of water-extractable substances between SP and AP. However, in the Shou Mei group, water-extractable substances were higher in SP than in AP. In summary, water-extractable substances did not show a trend of seasonal differences (Figure 3A). The content of the water-extractable substance showed a decreasing trend with increasing storage years, which is similar to previous studies [45].

#### 3.2.2. Free Amino Acid

Free amino acids are the basic units of protein and are organic compounds containing amine (-NH2) and carboxyl (-COOH) functional groups [47]. They are one of the most important energy sources for plants and are the precursors for the biosynthesis of many important secondary metabolites [10]. In this study, the free amino acid content of SPWTs was higher than that of APWTs, and this significant difference was observed in the 3-year samples (Figure 3B). Simultaneously, in the sensory evaluation results, the SPWTs scored higher than the APWTs in umami (Figure 2A,B). Free amino acids endow tea infusions with umami and smooth tastes, are the main contributors to the umami taste, and participate in the formation of aroma [48,49]. This may cause the umami taste and smoothness of SPWTs to be stronger than those of APWTs. Liu et al. [50] found that the nitrogen content stored in spring shoots is higher than in autumn shoots, providing an abundant source of nitrogen for amino acid synthesis. This may account for the difference in free amino acid content between SPWTs and APWTs. Additionally, with an increase in storage time, the content of free amino acids gradually decreased and was the highest in the samples with the shortest storage years, which was also reflected in the sensory evaluation results. These results corresponded well with previous findings among teas from different seasons [31].

#### 3.2.3. Caffeine

Alkaloids are small molecule nitrogenous compounds that, in tea trees, include mainly purine bases, pyrimidine bases, and pyridine bases, among which purine bases are the main components of alkaloids [51]. Purine alkaloids have the same purine ring structure in tea trees, whereas purine bases are dominated by caffeine [52]. Caffeine stimulates the central nervous system and enhances mental and physical processes in the human body [53]. It accounts for 3% of the dry weight of tea leaves and is one of the most important factors in determining the quality of green tea, with a bitter taste that enhances the bitterness and astringency of the tea infusion [54,55]. The caffeine content of SPWTs was higher than that of APWTs (Figure 3C). This may be due to the higher nitrogen content stored in spring shoots than in autumn shoots, which promotes photosynthetic carbon assimilation, consumes more photosynthetic products, then facilitates the synthesis and accumulation of caffeine [56]. This may result in a stronger astringency and thickness in SPWTs than in APWTs. The caffeine content does not show a significant trend of increase or decrease with increasing storage time, which was due to the stable structure of the purine ring of caffeine [33].

#### 3.2.4. Total Polyphenols

Polyphenols are compounds with one phenolic ring (phenolic acid/phenolic alcohol) or multiple aromatic rings with one or more hydroxyl groups and have been widely studied [57,58]. In our research, the total content of polyphenols in the white peony and Shou Mei tea samples from the same year was higher in SP than that in AP samples (Figure 3D), and with the increase in time, the content showed a downward trend, which is similar to the findings of previous studies [45]. The total polyphenol content of the same white tea was significantly different between SPWTs and APWTs of the same year. Tea polyphenols are associated with bitterness in sensory evaluation and enhancement of the astringency of tea infusions, which may be one reason why SPWTs have stronger astringency and thickness than APWTs [8]. The polyphenol content of tea tends to decrease as storage time increases. Technically, the decrease in tea polyphenols content with storage age may be due to the Folin-Ciocalteu reagent not reacting with the phenolic hydroxyl groups of polymeric polyphenols (e.g., TFs, TRs, and TBs) [45].

#### 3.2.5. Total Flavonoids

Flavonoids are a large class of structurally diverse analytes composed of a variety of basic skeletons and a series of derivatives. They are also the most representative secondary metabolites in tea and include flavonoids, flavonols, flavanones, flavanols, and anthocyanins [59]. In this study, the flavonoid content of APWTs was higher than that of SPWTs (Figure 3E), and the difference between SP and AP samples was significant. Zhang et al. [60] found the content of flavonoids increased at a higher temperature under the same growth conditions. Additionally, by checking historical temperatures, the average temperature in autumn is higher than that in spring, which causes the content of flavonoids to be higher in autumn than in spring (Appendix A). Tea flavonoids have attracted extensive attention because of their multiple roles in improving the resistance of fresh tea leaves to multiple stresses and forming unique flavours and colours in tea infusions, which enhance the bitterness of tea infusions [9,59]. This may result in a darker colour in APWTs than in SPWTs (Figure 1).

#### 3.2.6. Soluble Sugar

The soluble sugars in tea are mainly sucrose, followed by glucose and fructose, and starch hydrolysis is an important source [61,62]. Soluble sugars are the main taste compound in tea and bring distinctive sweetness to the tea infusion, which positively impacts the flavour [63,64]. In the present study, the soluble sugar content did not show seasonal differences and may not be an important factor in differentiating SPWTs and APWTs (Figure 3F).

#### 3.2.7. TFs, TRs, and TBs

TFs, TRs, and TBs are a range of catechin derivatives, which are also the main water-soluble pigments in tea infusions and affect their taste. TFs are compounds with a benzotropinone structure that are golden yellow in colour and pungent and astringent in flavour. They are an important component of the strength of taste and crispness of the tea infusion and are related to astringency [65]. TRs are complex bronzing phenols with a sweet, mellow flavour, which is an important component for the consistency and strength of the tea broth. TBs are dark brown macromolecular compounds, produced mainly by the oxidation of TRs and TFs, and they are negatively correlated with black tea infusions’ colour and taste [66,67,68]. Here, we found that the TF (Figure 3G) and TR (Figure 3H) contents were higher in SPWTs than in APWTs in the first two years, but the difference was not significant in the third year. The TB (Figure 3I) content was lower in SPWTs than in APWTs in the latter two years, whereas the difference was not significant in the first year. The combined effects of TFs, TRs, and TBs may contribute to differences in the production of SPWTs and APWTs. The content of TBs showed a general increasing trend with the decrease in tea polyphenols, suggesting that increased storage time enhances the polymerisation of tea polyphenols and leads to an increase in the content of TBs with storage. This result was in accordance with the findings by Zhao et al. [45]. 

#### 3.2.8. Free Amino Acids, Tea Polyphenols, Caffeine, Flavonoids, and Theaflavin May Be the Important Macro-Composition That Distinguish SPWTs and APWTs

To further identify the key macro-components that differentiate the tastes of SPWTs and APWTs, OPLS-DA was performed on the macro-compositions (Figure 4). The fitted parameters are R2Y = 0.841 and Q2 = 0.826. The permutation test (Figure 4B) showed that Q2 = 0.954, R2Y = 0.964, and R2X = 0.82, with a mean *p*-value of 0.005 for Q2 and R2, indicating that the OPLS-DA discriminant model was not overfitted and was relatively reliable. SPWTs gathered in the first and fourth quadrants, and APWTs were clustered in the second and third quadrants. The indication of the characteristic macro-composition helps to distinguish SPWTs from APWTs (Figure 4A). It can be seen from the S-plot (Figure 4C) that the variable importance in projection (VIP) value of the red point is >1, and metabolites with VIP > 1 are generally considered to be significantly different. Thus, free amino acids, tea polyphenols, caffeine, flavonoids, and TFs were the important macro-components that distinguished SPWTs from APWTs.

### 3.3. Identification of VOCs in SPWTs and APWTs

#### A Total of 545 Volatiles Were Detected

We used HS-SPME in combination with GC-MS/MS to determine the volatile content of white tea during different seasons. Total ion flow (TIC) plots of the mixed samples were superimposed to determine the reproducibility of volatile extraction and detection. The high stability of the instrument ensured reproducible and reliable data (Appendix A).

In total, 545 volatiles were detected, including 116 terpenes, 87 hydrocarbons, 83 esters, 69 alcohols, 63 heterocyclic compounds, 52 aldehydes, 46 ketones, 14 amines, 9 acids, 7 phenols, 6 aromatics, 4 halogenated hydrocarbons, and 3 ethers (Appendix A). Correlation analyses were performed to assess biological replicates within each group. The results showed good reproducibility for three samples in each group (Appendix A). Overall, the composition of VOCs in SPWTs and APWTs was similar, but the exact content differed slightly (Appendix A). Most VOCs were hydrocarbons (average of 26.46%), followed by heterocyclic compounds (average of 22.98%), and terpenes (average of 22.58%), with the total content of hydrocarbons, heterocyclic compounds, and terpenes exceeding 60%. Although significant amounts of esters, alcohols, ketones, and aldehydes were observed, their content did not exceed 9% in any group of white teas. Acid and ether contents ranged from 0.94% to 2.31% and were low in phenols, aromatics, and halogenated hydrocarbon compounds.

Of all the VOCs, the top ten detected substances in relative terms were 3-propylpyridine, 2,6-dimethylpiazine, 5-ethyl-2(5H)-furanone, 2-isopropyl-5-methylpyrazine, 5-methyl-(E)-2-hepten-4-one, 4-methyl-1-(1-methylethyl)-bicyclo [3.1.0]hex-2-ene, 2,6,6-trimethyl-octane, bicyclo(3.3.1)non-2-ene, sabinene, and α-fenchene. The relative content of 4-methyl-1-(1-methylethyl)-bicyclo[3.1.0]hex-2-ene in SPWTSs was the highest, with an average proportion of 8.53%, followed by sabinene (8.51%), octane, 2,6,6-trimethyl- (7.89%), and 3-propylpyridine (5.62%), which mainly displayed woody or sweet aromas [58]. However, 3-propylpyridine had the highest relative content in APWTs, with an average proportion of 14.42%, followed by 4-methyl-1-(1-methylethyl)-bicyclo[3.1.0]hex-2-ene (8.05%), sabinen (8.03%), and octane, 2,6,6-trimethyl- (7.48%). The higher relative contents of these VOCs may result in a stronger aroma in SPWTs than in APWTs.

### 3.4. Screening for Differential VOCs in SPWTs and APWTs

#### 3.4.1. PCA of VOCs in SPWTs and APWTs

The PCA based on all the identified VOCs was performed to analyze the confidence of the identification results and the overall compound differences among the 12 groups (Figure 5A). The first two principal components (PCs) accounted for 55.09% of the total variation (PC1 = 30.19%, PC2 = 24.9%). The SPWTs were distributed on the positive side of the first component, and the APWTs were distributed on the negative side of the first component. CSM19 and QSM19 were relatively independent of each other in the PC2 direction. Observing the loading plot (Figure 5B), a total of 189 VOCs were in the third quadrant, and the markers a, b, c, d, and e contributed greatly to the differences between groups, namely (1R,4S,5S)-1,8-dimethyl-4-prop-1-en-2-yl-spiro[4.5]dec-8-ene, p-Tolyl isobutyrate, 1-(Furan-2-yl)-2-methylpentan-1-one, 4-(1-methylethyl)-Benzaldehyde, and 2,3,6-trimethyl phenol. P-Tolyl isobutyrate has a “green” and “floral” aroma; 4-(1-methylethyl)-Benzaldehyde has a “green” and “herbal” aroma. The difference in their content between sample groups may be responsible for the stronger “fresh” and “flower” aromas of CSM19 than QSM19. Overall, the samples were well-differentiated based on the classification of seasons and were clustered using three biological replicates, indicating that the volatile content in different seasons varied greatly.

#### 3.4.2. OPLS-DA of VOCs in SPWTs and APWTs

To investigate the VOCs responsible for the sensory differences between SPWTs and APWTs, we used OPLS-DA for further analysis. Compared with PCA, OPLS-DA provides a supervised classification that can more effectively differentiate samples and extract information between different variables [69]. OPLS-DA was used to maximise the separation of samples and to better differentiate the main differences between SPWTs and APWTs. The OPLS-DA model was constructed based on 545 volatiles (Figure 6A), with a fit parameter of R^2^Y = 0.975 and Q^2^ = 0.945. The cross-validation model was tested 200 times (Figure 6B), and the intercept of the Q^2^ regression line with the Y-axis was less than 0, indicating that the OPLS-DA discriminant model was not overfitted and that the model was reliable (R^2^ = 0.377, Q^2^ = −0.758). SPWTs were clustered in Quadrants 1 and 4, and APWTs were clustered in Quadrants 2 and 3, with SPWTs and APWTs being significantly differentiated by characteristic volatiles.

There were 54 compounds with VIP values greater than 1 in the OPLS-DA model that was developed in this study. To make the results more accurate, compounds with VIP values greater than 1 were further analyzed using the Kruskal-Wallis test [70]. Among the 54 compounds with VIP values > 1, the *p*-values of all compounds were <0.05 (Appendix A), indicating that there were statistical differences between white tea samples in different picking seasons, which were important aromatic components of white tea in different seasons.

Further analysis of the 54 VOCs showed that terpenoids and heterocyclic compounds dominated the numbers, followed by aldehydes and esters. Terpene alcohols have characteristics of “flowery” and “fruit” aromas and are one of the most important compounds influencing the aroma formation of tea leaves [71,72,73]. Among the terpenoids with VIP values >1, sabinen had the highest VIP value, followed by (E)-4,8-dimethylnona-1,3,7-triene, α-fenchene, and γ-terpinene. The aromas are mostly “woody”; of these, sabinen and γ-terpinene were the most abundant aromatic components of white peony teas [74,75]. The sabinen relative content in SPWTs (25,246.65 μg kg^−1^ on average, relative to internal standard) was almost twice that of the content in APWTs (11,887.09 μg kg^−1^ on average, relative to internal standard).

During the drying process, high temperatures results in a Maillard reaction, which causes polymerisation and condensation of amino acids and carbonyl compounds at room temperature or when heated, resulting in a large number of heterocyclic compounds, such as pyrazines, pyrroles, pyridines, indoles, and furans [23,76]. In this study, the free amino acid content of SPWTs was higher than that of APWTs, which may have led to a higher content of heterocyclic compounds in spring than in autumn. 2-isopropyl-5-methylpyrazine, 2,6-dimethylpiazine, 3-propylpyridine, and 5-ethyl-2(5h)-furanone are heterocyclic compounds with VIP values > 1. They provide mostly nutty, roasted, and caramelized aromas [77]. In this study, 2-acetylpyrrole, 3-ethyl-2,5-dimethylpyrazine, and 3, 5-dimethyl-2-ethylpyrazine were detected. 2-acetylpyrrole was reported to have a high roasting aroma in Laoshan green tea; 3-ethyl-2,5-dimethylpyrazine and 3,5-dimethyl-2-ethylpyrazine are the main aromatic compounds in Wuyi rock tea and baked goods, whereas 3,5-dimethyl-2-ethylpyrazine contributes to nutty aromas in high-grade Japanese sencha and Chinese Longjing teas [78,79,80].

Aldehydes have a notably lower odour threshold and play an important role in the overall aroma formation of tea samples [81]. The 2-hexenal is associated with “green” and “grassy” flavours in black tea leaves, and (E)-2-hexenal is one of the main VOCs in white tea and is considered a characteristic aromatic component of Shou Mei tea [82,83]. Esters, especially those with a pleasant aroma, have been identified as key compounds in the formation of the characteristic aroma of white tea [84]. Of the six esters with a VIP greater than 1, 2-methylpropyl butanoate, caramel furanone, and ethyl 3-Hydroxyhexanoate were higher in SPWTs than in APWTs, all with “sweet” and “fruity” aromas.

The aromatic characteristics of the 54 volatile components were described differently, including terms such as “fruity”, “woody”, “floral”, “fresh”, and “sweet”. It is hypothesized that the difference in the content of these compounds in SPWTs and APWTs results in stronger “flower” and “fresh” aromas in SPWTs than in APWTs.

#### 3.4.3. OAV of VOCs in SPWTs and APWTs

The contribution of VOCs to the aroma characteristics of SPWTs and APWTs cannot be determined only by the content of VOCs, and the contribution of the volatiles to the overall aroma of the tea depends on the quantitative abundance of the compounds and the odour threshold [85,86,87]. Therefore, the OAV was introduced to further explore the key VOCs of SPWTs and APWTs. Important aromatic compounds with OAV >1 were selected for analysis by querying the compounds for their water thresholds and calculating their OAV values. Nine aromatic compounds were obtained (Figure 7 and Appendix A): sabinen, trans-2-hexenal, 2-hexenal, (2E,4E)-2,4-octadienal, β-pinene, (-)-cis-rose oxide, caramel furanone, trans-rose oxide and rose oxide. Among them, the relative contents of six aromatic compounds, sabinen, (2E,4E)-2,4-octadienal, (-)-cis-rose oxide, trans-rose oxide, rose oxide, and caramel furanone were significantly different between spring and autumn in the same year. Rose oxides can be detected in various essential oils and contribute significantly to the varietal aroma of Gewürztraminer wines [88,89,90]. Rose oxide, trans-rose oxide, and (-)-cis-rose oxide all have “rose”, “floral”, and “green” aromas, their relative content increases with the increase in sample years, and the seasonal difference is significant. Among that group, (-)-cis-rose oxide was detected in QMD19 but not in other autumn groups, which may be the reason for the stronger “flower” aroma in the white peony tea samples in SP (Figure 2) [91,92]. Caramel furanone had the highest relative content in CMD19 (709.7 ± 28.08 μg kg^−1^, relative to internal standard), and the relative content in SP samples was 6–22 times that of the content in AP samples. Caramel furanone is one of the main aromatic substances in the caramel-like aromas of Hanzhong black tea [93]. It is noteworthy that the relative contents of rose oxide, trans-rose oxide, (-)-cis-rose oxide, (2E,4E)-2,4-octadienal, and caramel furanone show a similar pattern in the samples. Importantly, (2E,4E)-2,4-octadienal had an OAV > 1 in all spring samples and < 1 in all autumn samples. (-)-cis-rose oxide had an OAV < 1 in all autumn samples, except for OAV > 1 in QMD19. These results show that among the nine types of aromatic compounds screened, six are characteristic aromatic compounds that distinguish SPWTs from APWTs, and (2E,4E)-2,4-octadienal, (-)-cis-rose oxide is the key compound.

## 4. Conclusions

In this study, sensory evaluation, biochemical composition determination, GC-MS, OAV analysis, and multivariate statistical analysis were combined to measure the macro-compositions and VOCs of SPWTs and APWTs. The “umami”, “smoothness”, “thickness”, and “sweetness” tastes as well as the “flower” and “fresh” aromas were stronger in SPWTs than in APWTs. The contents of free amino acids, caffeine, and tea polyphenols in SPWTs were higher than those in APWTs, while the flavonoids content was lower than APWTs. The results of OPLS-DA showed that free amino acids, tea polyphenols, caffeine, flavonoids, and TFs were important macro-components for distinguishing SPWTs from APWTs. The difference in their content leads to a difference in taste between SPWTs and APWTs. A total of 545 VOCs were detected by GC-MS, and the key compounds were screened by stoichiometry, OAV analysis, and OPLS-DA analysis. The results indicated that (2E,4E)-2,4-octadienal and (-)-cis-rose oxide were the key VOCs that distinguished SPWTs from APWTs. The aroma of (2E,4E)-2,4-octadienal is described as “green”, and the aroma of (-)-cis-rose oxide is described as “rose”, “green”, and “floral”. Their higher content in SPWTs than APWTs is responsible for the stronger “flower” and “fresh” aroma of SPWTs than APWTs. In the future, the molecular regulatory mechanisms underlying the quality differences between SPWTs and APWTs will be further explored. This study contributes to an in-depth understanding of the taste and aroma of white tea and provides a scientific basis for future work to improve the quality of white tea.

## Figures and Tables

**Figure 1 foods-11-03628-f001:**
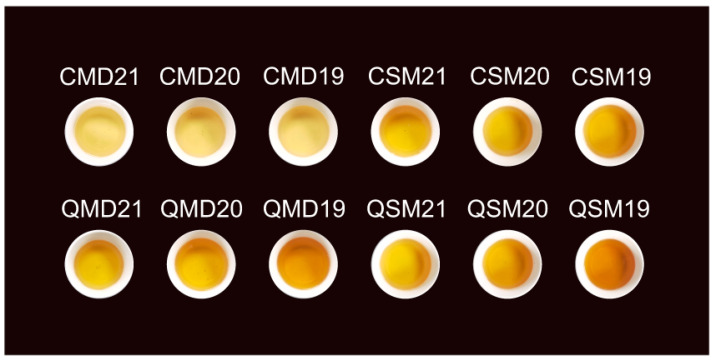
The tea infusions of SPWTs and APWTs leaves. White teas picked in spring were called SPWTs and those picked in autumn were called APWTs. White peony teas and Shou Mei picked in spring were termed CMD19-21 and CSM19-21, respectively, while those picked in autumn were correspondingly named QMD19-21 and QSM19-21. The number following CMD, QMD, CSM, and QSM represented the production years, corresponding to 2019, 2020, and 2021, respectively (Table 1).

**Figure 2 foods-11-03628-f002:**
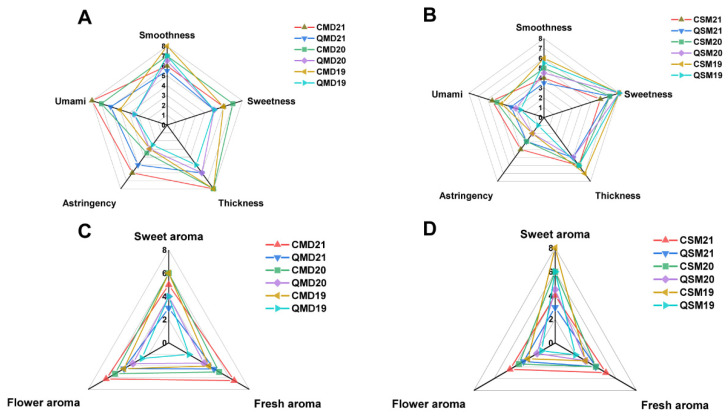
Radar map of the taste and aroma of SP and AP white peony (**A**,**C**) and Shou Mei (**B**,**D**) teas. White teas picked in spring were called SPWTs and those picked in autumn were called APWTs. White peony teas and Shou Mei picked in spring were termed CMD19-21 and CSM19-21, respectively, while those picked in autumn were correspondingly named QMD19-21 and QSM19-21. The number following CMD, QMD, CSM, and QSM represented the production years, corresponding to 2019, 2020, and 2021, respectively (Table 1).

**Figure 3 foods-11-03628-f003:**
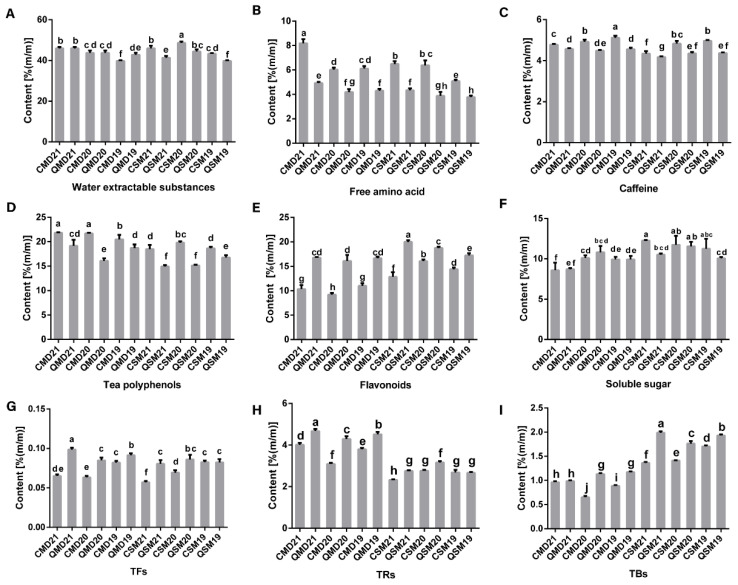
The content of (**A**) water-extractable substances, (**B**) free amino acids, (**C**) caffeine, (**D**) tea polyphenols, (**E**) flavonoids, (**F**) soluble sugar, (**G**) TFs, (**H**) TRs, and (**I**) TBs. The various superscripts show significant differences (*p* < 0.05). White teas picked in spring were called SPWTs and those picked in autumn were called APWTs. White peony teas and Shou Mei picked in spring were termed CMD19-21 and CSM19-21, respectively, while those picked in autumn were correspondingly named QMD19-21 and QSM19-21. The number following CMD, QMD, CSM, and QSM represented the production years, corresponding to 2019, 2020, and 2021, respectively (Table 1).

**Figure 4 foods-11-03628-f004:**
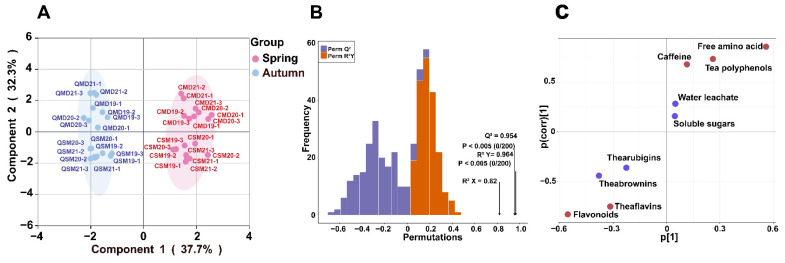
(**A**) OPLS-DA scores scatter plot of non-volatile components; (**B**) OPLS-DA cross-validation result of nonvolatile components; and (**C**) OPLS-plot result. In Figure 4C, the red dots indicate that the VIP value is greater than 1, and the blue dots indicate that the VIP value is less than 1. White teas picked in spring were called SPWTs and those picked in autumn were called APWTs. White peony teas and Shou Mei picked in spring were termed CMD19-21 and CSM19-21, respectively, while those picked in autumn were correspondingly named QMD19-21 and QSM19-21. The number following CMD, QMD, CSM, and QSM represented the production years, corresponding to 2019, 2020, and 2021, respectively. The number after the year indicates a biological duplicate (Table 1).

**Figure 5 foods-11-03628-f005:**
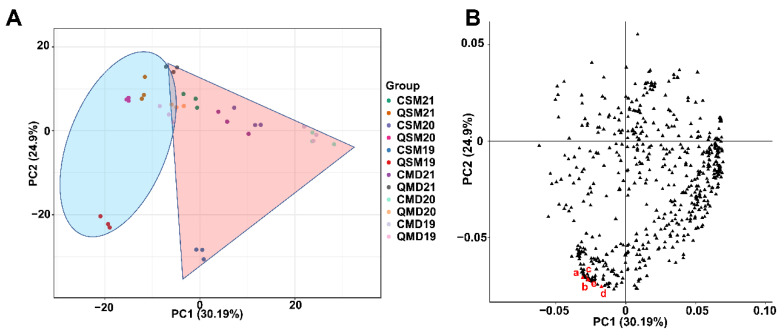
(**A**) PCA of VOCs in SPWTs and APWTs; and (**B**) PCA loading score plots. The points falling in the shaded area of the red triangle are SPWTs and in the shaded area of the blue oval are APWTs. In Figure 5B, triangles indicate all detected VOCs, with the top five VOCs contributing most to the difference between groups marked in red and the others in black. White teas picked in spring were called SPWTs and those picked in autumn were called APWTs. White peony teas and Shou Mei picked in spring were termed CMD19-21 and CSM19-21, respectively, while those picked in autumn were correspondingly named QMD19-21 and QSM19-21. The number following CMD, QMD, CSM, and QSM represented the production years, corresponding to 2019, 2020, and 2021, respectively (Table 1).

**Figure 6 foods-11-03628-f006:**
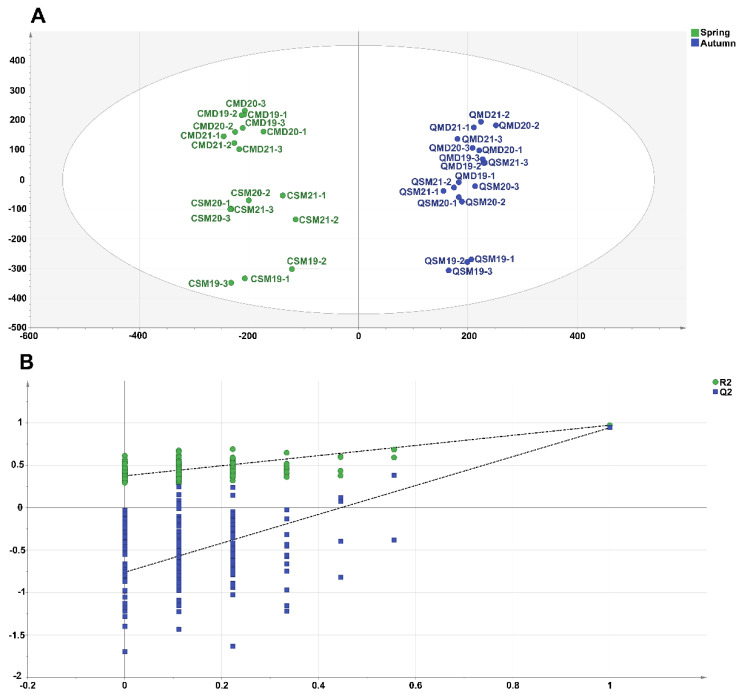
(**A**) OPLS-DA scores scatter plot of volatile components; and (**B**) OPLS-DA cross-validation result of VOCs. White teas picked in spring were called SPWTs and those picked in autumn were called APWTs. White peony teas and Shou Mei picked in spring were termed CMD19-21 and CSM19-21, respectively, while those picked in autumn were correspondingly named QMD19-21 and QSM19-21. The number following CMD, QMD, CSM, and QSM represented the production years, corresponding to 2019, 2020, and 2021, respectively. The number after the year indicates a biological duplicate (Table 1).

**Figure 7 foods-11-03628-f007:**
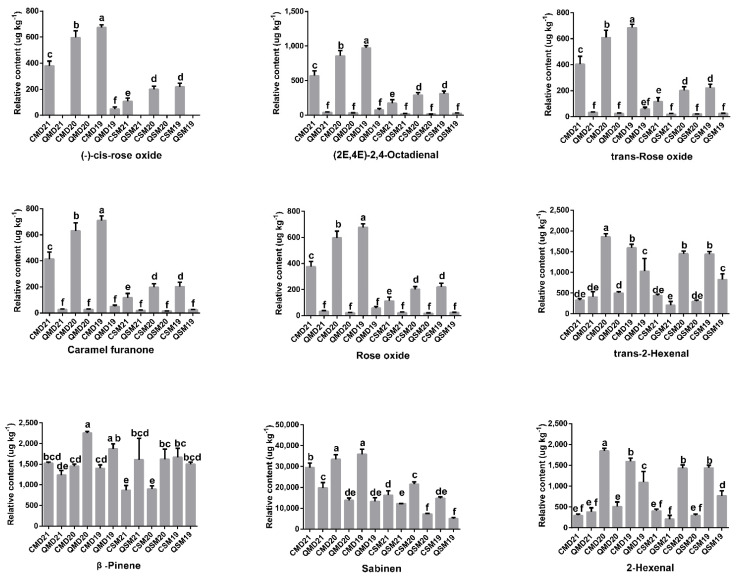
The relative content of nine aromatic compounds which had OAV > 1. The different superscripts show significant differences (*p* < 0.05). White teas picked in spring were called SPWTs and those picked in autumn were called APWTs. White peony teas and Shou Mei picked in spring were termed CMD19-21 and CSM19-21, respectively, while those picked in autumn were correspondingly named QMD19-21 and QSM19-21. The number following CMD, QMD, CSM, and QSM represented the production years, corresponding to 2019, 2020, and 2021, respectively (Table 1).

**Table 1 foods-11-03628-t001:** White tea sample information.

White Tea	Picked Season of Leaves	Identifier	Production Date	Picking Standards	Identifier
White peony tea	Spring	SPWT	2021.4–2021.5	one bud with the first to second tender leaves	CMD21-1 to 3
			2020.4–2020.5		CMD20-1 to 3
			2019.4–2019.5		CMD19-1 to 3
	Autumn	APWT	2021.9–2021.10		QMD21-1 to 3
			2020.9–2020.10		QMD20-1 to 3
			2019.9–2019.10		QMD19-1 to 3
Shou Mei	Spring	SPWT	2021.4–2021.5	one bud with the third to fifth leaves	CSM21-1 to 3
			2020.4–2020.5		CSM20-1 to 3
			2019.4–2019.5		CSM19-1 to 3
	Autumn	APWT	2021.9–2021.10		QSM21-1 to 3
			2020.9–2020.10		QSM20-1 to 3
			2019.9–2019.10		QSM19-1 to 3

## Data Availability

Data is contained within the article or Appendix A.

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
