# Peer review of "A Comprehensive Investigation of Macro-Composition and Volatile Compounds in Spring-Picked and Autumn-Picked White Tea"

_foods, 2022, doi:10.3390/foods11223628_

Round 1
Reviewer 1 Report
Dear Authors,
I reviewed the manuscript entitled “A Comprehensive Investigation of Macro-Composition and Volatile Compounds in Spring-Picked and Autumn-Picked White Tea” submitted to the journal Foods. This study compared white teas harvested in 2 different conditions : in spring and in autumn in order to evaluate the impact of harvest time on the chemical composition and sensory perception of the tea infusions. The investigation has been conducted on 2 different tea grades, and over 3 years to assess the effect of storage time. Teas were produced from the same manufacturer, which aimed to standardize their processing (and thus eliminate related biases). The non-volatile composition in polyphenols, caffeine, free amino acids and sugars has been conducted using different assay, whereas the volatile compounds were investigated using a non-targeted approach based on gas chromatography. The sensory analysis has been performed thanks to an expert panel. The results showed that the 2 harvest times induced systematic differences in chemical composition and flavor. Key volatile compound have been determined to be key for differentiating the spring and autumn harvest.
General comment :
I believe that this study has been well conducted in terms of conditions, replicates, and methods. This interesting study aimed at filling the gap in literature regarding the effect of harvest time on white tea specifically as opposed to other tea types. The introduction was namely well written but could benefit from additional context information. However, there are a few elements that raise questions.
Firstly, some analytical procedure are cited from Chinese national standards that are not freely accessible. I ask that the procedure are briefly described or to cite an accessible document or a paper published in peer-reviewed journals. I also find relevant to include key information contained in Table S2 in the main text.
Also, I have found the results part difficult to read because the sample codes are never explained in the manuscript. It makes it difficult to tie them to the studied conditions and to read the figures. As they are explained as supplementary material, I request that Table S1 is included in the main text. The explanation of the sample codes should also be included in the figure caption. This should be corrected. Although I understand how each figure is used, I question the relevancy of including figure 5 in the main text. I believe that most of the figures are not sufficiently interpreted, leaving interesting details behind (please check my specific comments). Maybe the interpretation would benefit from defining a condition as control in order to structure the comparisons in text.
I would like to raise important points regarding the sensory part. The absence of statistical treatment makes it difficult to make definitive statements and I recommend to provide such analysis. Moreover, it is a shame that the olfactive sensory result was not interpreted in the light of the chemical analyses. I suggest that you develop this cross interpretation and change the order of the subsections. The sensory results section should be move after the VOC analysis section. As a result, the comments about non volatile substances and gustative perception could be made in this section. In turn, more extensive interpretation should be provided for the non-volatile analyses results.
The conclusion is too concise and fails to express how the results produced can be useful in tea production. As it is now, apparent interest of this research remains limited.
In the light of those arguments, I recommend major revision for this manuscript.
Specific comments:
Line 23: Please remove the capital letter to “sabinen”.
Line 46 : VOC stands for volatile olfactive compound. Please add the word olfactive for more coherence.
Lines 53-54: The link between molecular families and flavor needs to be further detailed. Applies for the lines 69-70 as well.
Lines 55-56: It could be useful to briefly describe the specific processing of a white tea as opposed to a green or black tea.
Line 61: You describe earlier that peony and Shou Mei are both harvested at 2 periods of time in the year. This sentence seems to contradict this. Please rephrase and clarify.
Line 65 : "Fresh" needs to be defined. Is it like the cold sensation of menthol ? Is it like fresh grass as opposed to dry grass ? This also applies to line 119. reference molecules could be mentioned to help.
Line 65: please explain how the aroma lasts more. Do you mean that it lasts more during the consumption in the tea cup ? I am familiar with "long finish" for gustative sensations, but not for olfactive sensations.
Line 66: From the introduction I understand that all tea types are impacted by the harvest season. Is there a reason why you specifically chose white tea over other types besides filling the void in the literature ?
Lines 85-88: This part does not belong to the introduction, more the abstract.
Lines 100-101: Put Camellia sinensis into italic please.
Line 101: Please precise what type of coordinate this is to clarify.
Line 119: Sweet taste is obvious but sweet aroma is not to me, please define.
Line 130-133: please elaborate on the methods even if papers are cited.
Line 139: Please add companies city and country.
Line 161-162: For R, please specify the version rather than the website.
Section title 3.1: Stating a result as sub-section title is original but not conventional, I let the editor make comments about it if needed.
Lines 173-174: Remove “for characterizing many product details”.
Line 183: Check the presence of the word “newness” in the sentence.
Line 184-185: Since no statistical treatment has been applied (and it could be), this statement is not scientifically valid.
Figure 3: please precise if %(m/v) or %(m/m) with information regarding the nature of the liquid or solid. It would be very useful to get and idea of the absolute content (for example in mg/L). Could it be included as a secondary y axis or could you mention a range in the main text?
Line 280: Please define the VIP value.
Lines 280-281: Are tea polyphenols and flavanoids + thearubigin + theabrownin + theaflavin anti correlated on the plot ? If yes, this is contradictory. More generally, please elaborate on how you interprete the plot.
Figure 4C : Please explain the colors in the figure caption please. Please try to improve the resolution.
Figure 5 : How do you interprete this figure? it is very rich yet the elements are small. It is very difficult to read due to the amount of compounds detected. I do not believe that this heatmap is useful as it is now. Maybe filtration of compounds could make it more readable.
Lines 303-304: please rephrase, especially the part “none of the white teas in each group 303 contained >9%.”
Line 313: I suggest to always mention “relative concentration” or “relative content” when it applies for the VOC analysis, because of the use of a internal standard and the absence of specific calibration.
Lines 324-325: The interpretation could be more detailed, starting with the position of replicate for each condition. Then comments could be made regarding the clustering. The positions of the CSM19 and QSM19 could be extensively discussed namely.
The vector or parameter plot could give insights into the VOC that are determinant for the positions of the samples in the PCA and thus link key VOCs to the harvest time or storage time.
Line 343: Please change to “p-value” in lower case.
Lines 348-350: I fail to understand the transition with the previous paragraph . Please rephrase.
Line 354: Is the concentration an equivalent of internal standard? If yes, this should be mentioned. This also applies to line 403.
Line 379: please erase “through”
Figure 7: Please try to improve the resolution.
Line 401: Change “except in” to “except that it was detected in”.
Line 402 : If you refer to another study, change to "in white peony tea". If you refer to your own data, it should be presented in the paper and the figure or table should be cited.
Lines 424-425: Sensory analysis is barely mentioned in the conclusion, if the sensory results were interpreted in relationship to chemical analysis, it could be easier to include them in the conclusion. Emphasis should as well be put on the implications of the results for white tea production.
Author Response
Dear reviewer:
Thank you very much for constructive suggestions and comments, which greatly improve our manuscript entitled “A Comprehensive Investigation of Macro-Composition and Volatile Compounds in Spring-Picked and Autumn-Picked White Tea” (Manuscript ID: foods-1987872). All of the changes to the manuscript are indicated in the text using track changes. We wish that our answers could be as complete as possible to the question of reviewer.
Reviewer:
I reviewed the manuscript entitled “A Comprehensive Investigation of Macro-Composition and Volatile Compounds in Spring-Picked and Autumn-Picked White Tea” submitted to the journal Foods. This study compared white teas harvested in 2 different conditions : in spring and in autumn in order to evaluate the impact of harvest time on the chemical composition and sensory perception of the tea infusions. The investigation has been conducted on 2 different tea grades, and over 3 years to assess the effect of storage time. Teas were produced from the same manufacturer, which aimed to standardize their processing (and thus eliminate related biases). The non-volatile composition in polyphenols, caffeine, free amino acids and sugars has been conducted using different assay, whereas the volatile compounds were investigated using a non-targeted approach based on gas chromatography. The sensory analysis has been performed thanks to an expert panel. The results showed that the 2 harvest times induced systematic differences in chemical composition and flavor. Key volatile compound have been determined to be key for differentiating the spring and autumn harvest.
General comment:
I believe that this study has been well conducted in terms of conditions, replicates, and methods. This interesting study aimed at filling the gap in literature regarding the effect of harvest time on white tea specifically as opposed to other tea types. The introduction was namely well written but could benefit from additional context information. However, there are a few elements that raise questions.
Firstly, some analytical procedure are cited from Chinese national standards that are not freely accessible. I ask that the procedure are briefly described or to cite an accessible document or a paper published in peer-reviewed journals. I also find relevant to include key information contained in Table S2 in the main text.
Author’s response: Thank you for your valuable comment. We have re-written this part according to the reviewer’s suggestion. In the revised manuscript, a brief description of the procedures for some of the analytical methods has been provided and the key information contained in Table S2 has been included in the text.
Details in manuscript are as follow:
(1) Polyphenols were quantified using the Colorimetric method with Folin- Ciocalteu reagent based on the National Standards of China (GB/T 8312-2018). Specifically, 0.2 g of tea was added to 4.5 mL of 70% methanol (preheated to 70 ℃), after stirring and cool, centrifuge at 3500 r/min for 10 min, transfer the supernatant to a 10 mL test tube. Take 1mL of the supernatant into a volumetric flask and add 5 mL of Foline-Ciocalteau reagent and 4 mL of 7.5% Na2CO3, and was shaken for 8 min and made volume with pure water. The absorbance of the mixture was measured at 765 nm.
(2) Caffeine was quantified using the UV spectrophotometric method based on the National Standard of China (GB/T 8312-2013). Firstly, a 2 mL tea infusion was mixed with 4 mL of 0.01 mol/L hydrochloric acid and 1 mL basic lead acetate solution in a 100 mL volumetric flask. After added up to volume with water and mixed, the solution was filtered. The 12.5 mL filtrate was taken into 25 mL volumetric flask and added with 0.05 mL of 4.5 mol/L sulfuric acid to eliminate residual lead ion in the solution. After added up to volume with water and shaken, the mixed solution was filtered. Absorbance of the second filtrate was measured spectrophotometrically at 274 nm.
(3) The total free amino acids were quantified using the colourimetric method based on the National Standard of China (GB/T 8314-2013), with minor modifications. A 1 mL tea infusion and 0.5mL phosphate buffered saline were put into a 25mL test tubes, then 0.5 mL ninhydrin waw also added. The tubes were placed in a boiling water bath for 10 min, and then immediately cooled to room temperature in an ice bath. Finally, the absorbances of the samples was measured spectrophotometrically at 570 nm.
(4)TFs、TRs、TBs
The total content of theabrownins (TBs), theaflavins (TFs), and thearubigins(TRs) were determined using a previously discussed method[41,42]. Briefly, a total of 3g of dried tea was extracted with 125 mL boiling distilled water in a boiling water bath for 10 min and filtered, followed by cooled to room temperature. 50 mL of tea filtrate was pipetted in 50 mL of ethyl acetate with shaken for 5 min and the layers were separated after equilibration. A 0.8 mL portions of the ethyl acetate layer was made volume to 10 mL with 95% ethanol (solution A). Another 15 mL of the ethyl acetate layer was shaken with 15 mL of NaHCO3 solution (2.5%) for 30 s, stand until the layers separated. Pipetted 1.6 mL of ethyl acetate layer and fix the volume to 10 mL with 95% ethanol (solution C). A 0.8 mL sample of portions of the aqueous layer was diluted to 10 mL with 2.4 mL distilled water and 0.8 mL saturated oxalic acid solution (10.2%, m/v), and 6 mL of 95% ethanol (solution D). A 25 mL of tea filtrate was pipetted and mixed with 25 mL of butyl alcohol. After shaking for 3 min and the layers were separated after equilibration. A 0.4 mL sample of the aqueous layer (second) was made volume to 10 ml with 0.4 mL of the saturated oxalic acid solution, 1.2 mL of distilled water, and 95% ethanol (solution B). The absorbance of solutions A, B, C, and D at 380nm was measured spectrophotometrically using 95% ethanol as a blank. The results were calculated using the following formula:
TFs =100%
TRs = 100%
TBs= 100%
Volatiles were extracted from the samples using the headspace solid-phase microex-traction (HS-SPME) method with three replicates for each assay. After sampling, volatile analysis was performed using an Agilent Model 8890 GC equipped with a 30 m × 0.25 mm × 0.25 μm DB-5MS (5% phenyl-polymethylsiloxane) capillary column and a 7000D mass spectrometer (Agilent, Santa Clara, CA, USA). HS-SPME and gas chromatog-raphy-mass spectrometry (GC-MS) analysis procedures were performed as previously described, with minor modifications. The temperature was programmed at 40℃ for 3.5 min, increased at 10℃/min to 100℃; then increased at 7℃/min to 180℃, and finally in-creased at 25℃/min to 280℃, and held for 5 min. The flow rate of the carrier gas (helium, 99.999%) was 1.2 mL/min. The MS spectrometer was operated in electron impact mode with electron energy of 70 eV and a scan range of m/z 50–500. The ion source and mass spectrum transferline temperatures were 230 ◦C and 280 ◦C, respectively. The specific GC-MS acquisition conditions are listed in Supplementary Table S1.
Also, I have found the results part difficult to read because the sample codes are never explained in the manuscript. It makes it difficult to tie them to the studied conditions and to read the figures. As they are explained as supplementary material, I request that Table S1 is included in the main text. The explanation of the sample codes should also be included in the figure caption. This should be corrected. Although I understand how each figure is used, I question the relevancy of including figure 5 in the main text. I believe that most of the figures are not sufficiently interpreted, leaving interesting details behind (please check my specific comments). Maybe the interpretation would benefit from defining a condition as control in order to structure the comparisons in text.
Author’s response: Thank you for your valuable comment. We have provided a detailed description of the sample codes and added to the text as Table 1. In addition, we have added the description of the sample codes to the description of each figure. We have removed Figure 5 from the text and explained the data more fully.
Details in manuscript are as follow:
Spring-picked was referred to as SP, and autumn-picked were called AP. White teas picked in spring were called as SPWTs and those picked in autumn were called as APWTs. White peony teas and Shou Mei picked in spring were termed CMD19-21 and CSM19-21 respectively, while those picked in autumn were named QMD19-21 and QSM19-21. The number following CMD, QMD, CSM, and QSM represented the production years, corresponding to 2019, 2020, and 2021 respectively. The number after the year indicates a biological duplication (Table 1).
Table 1. White tea samples information
I would like to raise important points regarding the sensory part. The absence of statistical treatment makes it difficult to make definitive statements and I recommend to provide such analysis. Moreover, it is a shame that the olfactive sensory result was not interpreted in the light of the chemical analyses. I suggest that you develop this cross interpretation and change the order of the subsections. The sensory results section should be move after the VOC analysis section. As a result, the comments about non volatile substances and gustative perception could be made in this section. In turn, more extensive interpretation should be provided for the non-volatile analyses results.
Author’s response: Thank you for your valuable comment. We conducted a one-way ANOVA on the results of the sensory review and added the results of the analysis to the supplementary material (Supplementary Figure S1). In addition, we add VOCs to the analysis of olfactory sensory results and non-volatile compounds to the analysis of taste sensory results. We found differences in spring-picked and autumn-picked white teas during the sensory review, and in turn analyzed them for non-volatile compounds and VOCs to find differences. In the article, we have added correlations among non-volatile compounds and VOC and sensory review results, whereas changing the description order in the articles would make the discussion of chemical substances difficult. At the same time, this structure has been used in many articles (1-3). Therefore, we believe that keeping the original structure of the article was more appropriate for analyzing the differences in spring-picked and autumn-picked white teas.
Supplementary Figure S1
Reference
- Fan, F.; Huang, C.; Tong, Y.; Guo, H.; Zhou, S.; Ye, J.; Gong, S. Widely targeted metabolomics analysis of white peony teas with different storage time and association with sensory attributes. Food Chem. 2021, 362, 130257, doi: 10.1016/j.foodchem.2021.130257.
- Xiao, Z.; Wang, H.; Niu, Y.; Liu, Q.; Zhu, J.; Chen, H.; Ma, N. Characterization of aroma compositions in different Chinese congou black teas using GC-MS and GC-O combined with partial least squares regression. Flavour Frag. J. 2017, 32, 265-276, doi: 10.1002/ffj.3378.
- Ni, H.; Jiang, Q.; Zhang, T.; Huang, G.; Li, L.; Chen, F. Characterization of the Aroma of an Instant White Tea Dried by Freeze Drying. Molecules 2020, 25, 3628, doi: 10.3390/molecules25163628.
The conclusion is too concise and fails to express how the results produced can be useful in tea production. As it is now, apparent interest of this research remains limited.
Author’s response: Thank you for your valuable comment. In the conclusion, we have added a description of the chemical substance-sensory link, the direction of future research and the scientific basis for future work to improve the quality of white tea by contributing to a deeper understanding of the taste and aroma of white tea.
Details in manuscript are as follow:
In this study, sensory evaluation, biochemical composition determination, GC-MS, OAV analysis, and multivariate statistical analysis were combined to measure the macro-compositions and VOCs of SPWTs and APWTs. The “umami”, “smoothness”, “thickness”, “sweetness” taste, “flower” and “fresh” aromas were stronger in SPWTs than that in APWTs. The content of free amino acids, caffeine, and tea polyphenols in SPWTs was higher than that in APWTs, while the flavonoids content is lower than APWTs. The results of OPLS-DA showed that free amino acids, tea polyphenols, caffeine, flavonoids, and TFs were important macro-components for distinguishing SPWTs from APWTs. The difference in their content leads to a difference in taste between SPWTs and APWTs. A total of 545 VOCs were detected by GC-MS, and the key compounds were screened by stoichiometry, OAV analysis, and OPLS-DA analysis. The results indicated that (2E,4E)-2,4-octadienal and (-)-cis-rose oxide were the key VOCs that distinguished SPWTs from APWTs. The aroma of (2E,4E)-2,4-octadienal is described as "green" and the aroma of (-)-cis-rose oxide is described as "rose”, “green” and “floral". Their higher content in SPWTs than APWTs is responsible for the stronger “flower” and “fresh” aroma of SPWTs than APWTs. In the future, the molecular regulatory mechanisms underlying the quality differences between SPWTs and APWTs will be further explored. This study contributes to an in-depth understanding of the taste and aroma of white tea and provides a scientific basis for future work to improve the quality of white tea.
Line 23: Please remove the capital letter to “sabinen”.
Author’s response: We are sorry for our careless mistake. We have made changes in the manuscript.
Line 46: VOC stands for volatile olfactive compound. Please add the word olfactive for more coherence.
Author’s response: Thanks for your comment. We have made changes in the manuscript.
Lines 53-54: The link between molecular families and flavor needs to be further detailed. Applies for the lines 69-70 as well.
Author’s response: Thanks for your valuable comment. We have revised the manuscript to add a link between molecular families and flavour.
Details in manuscript are as follow:
The quality of tea is determined by various distinctive secondary metabolites, mainly belonging to polyphenols, flavonoids, free amino acids, and volatile olfactive compounds (VOCs). Tea polyphenols enhance the bitterness of tea infusions, flavonoids substantial tea the bitter taste, free amino acids are the source of the umami taste, and VOCs give the tea a wide variety of aroma qualities. The biosynthesis of secondary metabolites was affected by many factors such as tea species, climate environment, soil, manufacturing processes, and picking season. Among them, the picking season is one of the key factors affecting the content of secondary metabolites. For example, the content of theanine, kaempferol-glycoside epicatechin, and catechin, which in fresh leaves in spring-picked (SP) is higher than that in autumn-picked (AP). Black tea in SP has higher contents of caffeine and catechin than that in AP. The contents of (-)-epigallocatechin gallate (EGCG) and (-)-epicatechin gallate (ECG) in oolong tea in SP were lower than those in AP, while the differences in catechin gallate (CG) and total catechins (TC) contents were not obvious. The content of free amino acids in SP green tea is higher than that in AP green tea. While the content of flavonoids is lower than that of AP green tea. The different contents of these compounds in SP and AP teas may have contributed to the different tastes. Fifteen VOCs, including trans-β-ionone, nonanal, and dimethyl sulfide, were found to be the main VOCs that distinguished SP Xinyang Maojian green teas from AP teas. Linalool, β-damascenone, and benzeneacetaldehyde were considered to be the main VOCs that distinguish Yingde black teas. A total of 12 aroma substances, including cis-jasmone, benzyl alcohol, and (E)-2-octenal, are key differentiating compounds between green teas of different seasons. Linalool has a “floral” odour; nonanal and (E)-2-octenal imparted a “green, fatty, or tallow” note; cis-jasmone was considered to have a “characteristic floral jasmine” odour; and benzyl alcohol has a “burning” taste and a “faint aromatic” odour. The different content of these VOCs in tea picked in different seasons contributes to the distinctive aroma of tea. Therefore, the picking season is an important factor in the formation of the flavour quality of tea, which has a significant impact on the taste and aroma of tea.
Lines 55-56: It could be useful to briefly describe the specific processing of a white tea as opposed to a green or black tea.
Author’s response: Thanks for your valuable comment. We have made changes in the manuscript.
Details in manuscript are as follow:
White tea is one of the six major types of tea in China and originated in Fujian province and is made from tea leaves that have been subjected to a long withering and drying process. White tea is characterized by an umami and a slightly sweet taste along with a fresh and green odour. It has been widely recognized for its health-promoting properties, including antioxidant, hypoglycemic, and lipid-lowering. Due to its potential health benefits and unique flavour, have led to a growing market share and widespread scientific interest.
Line 61: You describe earlier that peony and Shou Mei are both harvested at 2 periods of time in the year. This sentence seems to contradict this. Please rephrase and clarify.
Author’s response: We are sorry for our careless mistake. We have re-written in this part. Details in manuscript are as follow:
The main difference between white peony and Shou Mei teas is the maturity of the harvested leaves.
Line 65: "Fresh" needs to be defined. Is it like the cold sensation of menthol? Is it like fresh grass as opposed to dry grass ? This also applies to line 119. reference molecules could be mentioned to help.
Author’s response: Thanks for your valuable comment. We have defined the descriptions of taste and aroma.
Details in manuscript are as follow:
"Smoothness" is a pleasant taste with a silky softness and a buttery smoothness. "Sweetness" is the taste connected with sugary foods. "Thickness" was defined as the pleasure of having a coating on the tongue. "Astringency" is the sensation of dryness and contraction of the tongue and soft palate. The "Umami" was a delicious flavour induced by several amino acids. “Flower” was defined as a blend of natural floral odour. “Sweet” is the smell of sucrose solution. “Fresh” is fresh hay-like aroma.
Line 65: please explain how the aroma lasts more. Do you mean that it lasts more during the consumption in the tea cup ? I am familiar with "long finish" for gustative sensations, but not for olfactive sensations.
Author’s response: Thanks for your valuable comment. We have re-written in this part.
Details in manuscript are as follow:
The taste of SPWTs is more umami, and the aroma in the review glass are more lasting at temperatures of 75°C, 45°C and at room temperature
Line 66: From the introduction I understand that all tea types are impacted by the harvest season. Is there a reason why you specifically chose white tea over other types besides filling the void in the literature?
Author’s response: Thank you for your opinion. First of all, we added the potential health benefits of white tea in the article, second, we mentioned in the manuscript that its market share and attention have been increasing in recent years, and third, to further clarify the reasons for the difference between spring and autumn white tea can better provide support for the actual production.
Details in manuscript are as follow:
White tea is characterized by an umami and a slightly sweet taste along with a fresh and green odour. It has been widely recognized for its health-promoting properties, including antioxidant, hypoglycemic, and lipid-lowering. Due to its potential health benefits and unique flavour, have led to a growing market share and widespread scientific interest.
With a deeper understanding of the reasons for the differences between SPWTs and APWTs, production can be better guided.
Lines 85-88: This part does not belong to the introduction, more the abstract.
Author’s response: Thanks for your valuable comment. We have re-written in this part.
Details in manuscript are as follow:
Macro-composition quantification, multivariate statistical analysis, and odour activity value (OAV) analysis were also performed. This study will provide an objective and ac-curate theoretical basis for differentiating SPWTs and APWTs.
Lines 100-101: Put Camellia sinensis into italic please.
Author’s response: We are sorry for our careless mistake. We have re-written in this part.
Line 101: Please precise what type of coordinate this is to clarify.
Author’s response: Thanks for your valuable comment. We have re-written in this part.
Details in manuscript are as follow:
Samples of 12 white teas were made from fresh leaves of clonally propagated Camellia sinensis ‘Fudingdahaocha’ grown in the same tea field of Fuding Minghai Tea Co., Ltd in Fuding County, China (119°59′E,27°19′N).
Line 119: Sweet taste is obvious but sweet aroma is not to me, please define.
Author’s response: Thanks for your valuable comment. We have re-written in this part.
Details in manuscript are as follow:
"Smoothness" is a pleasant taste with a silky softness and a buttery smoothness. "Sweetness" is the taste connected with sugary foods. "Thickness" was defined as the pleasure of having a coating on the tongue. "Astringency" is the sensation of dryness and contraction of the tongue and soft palate. "Umami" was a delicious flavour induced by several amino acids. “Flower” was defined as a blend of natural floral odour. “Sweet” is the smell of sucrose solution. “Fresh” is fresh hay-like aroma.
Line 130-133: please elaborate on the methods even if papers are cited.
Author’s response: Thanks for your valuable comment. We have re-written in this part.
Details in manuscript are as follow:
(1) Polyphenols were quantified using the Colorimetric method with Folin- Ciocalteu reagent based on the National Standards of China (GB/T 8312-2018). Specifically, 0.2 g of tea was added to 4.5 mL of 70% methanol (preheated to 70 ℃), after stirring and cool, centrifuge at 3500 r/min for 10 min, transfer the supernatant to a 10 mL test tube. Take 1mL of the su-pernatant into a volumetric flask and add 5 mL of Foline-Ciocalteau reagent and 4 mL of 7.5% Na2CO3, and was shaken for 8 min and made volume with pure water. The absorb-ance of the mixture was measured at 765 nm.
(2) Caffeine was quantified using the UV spectrophotometric method based on the National Standard of China (GB/T 8312-2013). Firstly, a 2 mL tea infusion was mixed with 4 mL of 0.01 mol/L hydrochloric acid and 1 mL basic lead acetate solution in a 100 mL volumetric flask. After added up to volume with water and mixed, the solution was filtered. The 12.5 mL filtrate was taken into 25 mL volumetric flask and added with 0.05 mL of 4.5 mol/L sulfuric acid to eliminate residual lead ion in the solution. After added up to volume with water and shaken, the mixed solution was filtered. Absorbance of the second filtrate was measured spectrophotometrically at 274 nm.
(3) The total free amino acids were quantified using the colourimetric method based on the National Standard of China (GB/T 8314-2013), with minor modifications. A 1 mL tea infusion and 0.5mL phosphate buffered saline were put into a 25mL test tubes, then 0.5 mL ninhydrin waw also added. The tubes were placed in a boiling water bath for 10 min, and then immediately cooled to room temperature in an ice bath. Finally, the absorbances of the samples was measured spectrophotometrically at 570 nm.
(4)TFs、TRs、TBs
The total content of theabrownins (TBs), theaflavins (TFs), and thearubigins(TRs) were determined using a previously discussed method. Briefly, a total of 3g of dried tea was extracted with 125 mL boiling distilled water in a boiling water bath for 10 min and filtered, followed by cooled to room temperature. 50 mL of tea filtrate was pipetted in 50 mL of ethyl acetate with shaken for 5 min and the layers were separated after equilibration. A 0.8 mL portions of the ethyl acetate layer was made volume to 10 mL with 95% ethanol (solution A). Another 15 mL of the ethyl acetate layer was shaken with 15 mL of NaHCO3 solution (2.5%) for 30 s, stand until the layers separated. Pipetted 1.6 mL of ethyl acetate layer and fix the volume to 10 mL with 95% ethanol (solution C). A 0.8 mL sample of portions of the aqueous layer was diluted to 10 mL with 2.4 mL distilled water and 0.8 mL saturated oxalic acid solution (10.2%, m/v), and 6 mL of 95% ethanol (solution D). A 25 mL of tea filtrate was pipetted and mixed with 25 mL of butyl alcohol. After shaking for 3 min and the layers were separated after equilibration. A 0.4 mL sample of the aqueous layer (second) was made volume to 10 ml with 0.4 mL of the saturated oxalic acid solution, 1.2 mL of distilled water, and 95% ethanol (solution B). The absorbance of solutions A, B, C, and D at 380nm was measured spectrophotometrically using 95% ethanol as a blank. The results were calculated using the following formula:
TFs =100%
TRs = 100%
TBs= 100%
Line 139: Please add companies city and country.
Author’s response: We are sorry for our careless mistake. We have re-written in this part.
Line 161-162: For R, please specify the version rather than the website.
Author’s response: We are sorry for our careless mistake. We have re-written in this part.
Lines 173-174: Remove “for characterizing many product details”.
Author’s response: We are sorry for our careless mistake. We have re-written in this part.
Line 183: Check the presence of the word “newness” in the sentence.
Author’s response: We are sorry for our careless mistake. We have re-written in this part.
Line 184-185: Since no statistical treatment has been applied (and it could be), this statement is not scientifically valid.
Author’s response: Thanks for your valuable comment. We have re-written in this part. The results of the significant difference analysis of the taste and aroma attribute scores was added in Supplementary file (Supplementary Figure S1).
Figure 3: please precise if %(m/v) or %(m/m) with information regarding the nature of the liquid or solid. It would be very useful to get and idea of the absolute content (for example in mg/L). Could it be included as a secondary y axis or could you mention a range in the main text?
Author’s response: We are sorry for our careless mistake. We have re-written in this part.
Line 280: Please define the VIP value.
Author’s response: We are sorry for our careless mistake. We have re-written in this part.
Details in manuscript are as follow:
Variable Importance in Projection (VIP)
Lines 280-281: Are tea polyphenols and flavanoids + thearubigin + theabrownin + theaflavin anti correlated on the plot ? If yes, this is contradictory. More generally, please elaborate on how you interprete the plot.
Author’s response: Thank you for your opinion. Catechin compounds are the main components of tea polyphenols in tea plants, accounting for about 70%-80% of the total amount of tea polyphenols. Therefore, catechin content plays a major role in the analysis results of OPLS-DA of tea polyphenols in different samples. TFs, TRs, and TBs on the other hand, are a relatively low component of the tea polyphenols and does not play a decisive role.
Reference
Cabrera C, Artacho R, Gimenez R. Beneficial effects of green tea-a review. Journal of the American College of Nutrition, 2006,25(2):79-99.
Figure 4C : Please explain the colors in the figure caption please. Please try to improve the resolution.
Author’s response: Thanks for your valuable comment. We have modified the colours in Figure 4C, as well as increasing its resolution, and explained the colours in the note
Details in manuscript are as follow:
In Figure 4C, the red dot indicate that the VIP values is greater than 1 and blue dot indicate that the VIP values is less than 1.
Figure 5 : How do you interprete this figure? it is very rich yet the elements are small. It is very difficult to read due to the amount of compounds detected. I do not believe that this heatmap is useful as it is now. Maybe filtration of compounds could make it more readable.
Author’s response: Thanks for your valuable comment. We have removed Figure 5.
Lines 303-304: please rephrase, especially the part “none of the white teas in each group 303 contained >9%.”
Author’s response: We are sorry for our careless mistake. We have re-written in this part.
Details in manuscript are as follow:
Although significant amounts of esters, alcohols, ketones and aldehydes were observed, while their content did not exceed 9% in each group of white teas.
Line 313: I suggest to always mention “relative concentration” or “relative content” when it applies for the VOC analysis, because of the use of a internal standard and the absence of specific calibration.
Author’s response: Thanks for your valuable comment. We have re-written in this part.
Lines 324-325: The interpretation could be more detailed, starting with the position of replicate for each condition. Then comments could be made regarding the clustering. The positions of the CSM19 and QSM19 could be extensively discussed namely.
The vector or parameter plot could give insights into the VOC that are determinant for the positions of the samples in the PCA and thus link key VOCs to the harvest time or storage time.
Author’s response: Thanks for your valuable comment. We have re-written in this part.
Details in manuscript are as follow:
The PCA based on all the identified VOCs was performed to analyse the confidence of the identification results and the overall compound differences between the 12 groups (Figure 6A). The first two principal components (PCs) accounted for 55.09% of the total variation (PC1=30.19%, PC2=24.9%). The SPWTs were distributed on the positive side of the first component and the APWTs were distributed on the negative side of the first com-ponent. CSM19 and QSM19 were relatively independent of each other in the PC2 direction. Observing the loading plot (Figure 6B), a total of 189 VOCs were in the third quadrant, and the markers a, b, c, d, and e contributed greatly to the differences between groups, namely (1R,4S,5S)-1,8-dimethyl-4-prop-1-en-2-yl-spiro[4.5]dec-8-ene, p-Tolyl isobutyrate, 1-(Furan-2-yl)-2-methylpentan-1-one, 4-(1-methylethyl)-Benzaldehyde, and 2,3,6-trimethyl phenol. P-Tolyl isobutyrate has a “green” and “floral” aroma; 4-(1-methylethyl)-Benzaldehyde has a “green” and “herbal” aroma. The difference in their content between sample groups may be responsible for the stronger “fresh”, and “flower” aroma of CSM19 than QSM19. Overall, the samples were well differentiated based on the classification of seasons and were clustered using three biological replicates, indicating that the volatile content in different seasons varied greatly.
Line 343: Please change to “p-value” in lower case.
Author’s response: We are sorry for our careless mistake. We have re-written in this part.
Lines 348-350: I fail to understand the transition with the previous paragraph . Please rephrase.
Author’s response: Thanks for your valuable comment. We have re-written in this part.
Details in manuscript are as follow:
Further analysis of the 54 VOCs showed that terpenoids and heterocyclic compounds dominated the numbers, followed by aldehydes and esters. Terpene alcohols have characteristics of “flowery” and “fruit” aromas, and are one of the most important compounds influencing the aroma formation of tea leaves.
Line 354: Is the concentration an equivalent of internal standard? If yes, this should be mentioned. This also applies to line 403.
Author’s response: Thanks for your valuable comment. In fact, in section 2.5, we describe the method of calculating the relative content of VOCs, at the same time, we have applied your suggestions to the manuscript.
Details in manuscript are as follow:
(25246.65 μg kg−1 on average, relative to internal standard)
(11887.09 μg kg−1 on average, relative to internal standard)
(709.7±28.08 μg kg−1, relative to internal standard)
Line 379: please erase “through”
Author’s response: We are sorry for our careless mistake. We have re-written in this part.
Figure 7: Please try to improve the resolution.
Author’s response: Thanks for your valuable comment. We have increased the resolution
Details in manuscript are as follow:
Line 401: Change “except in” to “except that it was detected in”.
Author’s response: We are sorry for our careless mistake. We have re-written in this part.
Line 402 : If you refer to another study, change to "in white peony tea". If you refer to your own data, it should be presented in the paper and the figure or table should be cited.
Author’s response: We are sorry for our mistake. We have re-written in this part. Note added to previous text and sensory review form added to supplementary document.
Details in manuscript are as follow:
Rose oxide, trans-rose oxide, and (-)-cis-rose oxide all have “rose”, “floral”, and “green” aromas, and their relative content increases with the increase in sample years and the seasonal difference is significant, among which (-)-cis-rose oxide, except that it was detected in QMD19, but not in other autumn groups, which may be the reason for the stronger “flower” aroma in the white peony tea samples in SP (Figure 2)
Lines 424-425: Sensory analysis is barely mentioned in the conclusion, if the sensory results were interpreted in relationship to chemical analysis, it could be easier to include them in the conclusion. Emphasis should as well be put on the implications of the results for white tea production.
Author’s response: Thanks for your valuable comment. We have re-written in this part.
Details in manuscript are as follow:
In this study, sensory evaluation, biochemical composition determination, GC-MS, OAV analysis, and multivariate statistical analysis were combined to measure the macro-compositions and VOCs of SPWTs and APWTs. The “umami”, “smoothness”, “thickness”, “sweetness” taste, “flower” and “fresh” aromas were stronger in SPWTs than that in APWTs. The content of free amino acids, caffeine and tea polyphenols in SPWTs were higher than that in APWTs, while the flavonoids content is lower than APWTs. The results of OPLS-DA showed that free amino acids, tea polyphenols, caffeine, flavonoids, and TFs were important macro-components for distinguishing SPWTs from APWTs. The difference in their content leads to a difference in taste between SPWTs and APWTs. A total of 545 VOCs were detected by GC-MS, and the key compounds were screened by stoichiometry, OAV analysis, and OPLS-DA analysis. The results indicated that (2E,4E)-2,4-octadienal and (-)-cis-rose oxide were the key VOCs that distinguished SPWTs from APWTs. The aroma of (2E,4E)-2,4-octadienal is described as "green" and the aroma of (-)-cis-rose oxide is described as "rose”, “green” and “floral". Their higher content in SPWTs than APWTs is responsible for the stronger “flower” and “fresh” aroma of SPWTs than APWTs. In the future, the molecular regulatory mechanisms underlying the quality differences between SPWT and APWT will be further explored. This study contributes to an in-depth understanding of the taste and aroma of white tea, and provides a scientific basis for future work to improve the quality of white tea.

Reviewer 2 Report
Introduction
In the first paragraph, the authors reported the current state of knowledge on the differences between spring and autumn tea, pointing out that scientists have already proven that the timing of tea harvesting affects the qualitative and quantitative profile of the compounds it contains. The authors then moved on to introduce the news of white tea, which originated in China.
The last paragraph I consider the most important crowning the introduction of the article. The authors describe the purpose and scope of the paper while pointing out the procedures performed to compare the data collected in three consecutive years.
Materials and Methods
line 106 - I propose to elaborate one again here the abbreviations SPWT and APWT, because the authors introduce the research material and the reader should know what it is without reading the introduction.
line 114-117 - And what was done with the buds and leaves after the brewing was finished? were discarded or did they remain in the brew? this needs to be specified in the text
2.4. Macro-composition quantification – lines 124-133 - the authors must at least briefly describe the methodologies for analyzing each group of compounds. Chinese standards are not widely available in a language other than Chinese therefore this chapter does not allow other researchers to reproduce the methodology. Also, citing the methodology from another article is insufficient and the methodology should be at least briefly described.
2.5. Identification and analysis of volatiles - how the authors identified the various volatile compounds? please write in the methodology
Results and Discussion
3.2. Analysis of macro-composition of SPWTs and APWTs - the authors described the results of the work very poorly - mostly in the form of 1-3 sentences for each analysis. Most of the text is definitions and description of knowledge from the literature. There is also no explanation as to why any group of compounds was more or less in the infusions, and there is no discussion of the results with other studies.
Figure 3 - The authors did not specify which year of the 2019-2021 period these results come from.
Figure 4 A,B,C - The images/diagrams are unreadable, nothing can be seen on them.
Figure 5 - I recommend removing the heat map figure because it is very unreadable and adds little information to the article.
3.4.1. PCA of VOCs in SPWTs and APWTs - The authors did not describe in detail what the chart shows, what the classification looks like. Are there any regularities about the timing or years of harvesting?
line 356- Merad reaction? what is this?
Figure 7 - Like the other charts it is poorly readable and if there were no title it would be impossible to know what it represents.
Conclusion
line 418-420- this is a repetition of information already given earlier therefore I recommend removing this text.
The authors made a great deal of determinations but gave very few conclusions. There is no reference to the seasonality of the harvest and its impact on the formation of non-volatile compounds. In addition, they did not state how spring and autumn teas differ (volatile and non-volatile compounds). They did not state what needs to be followed to distinguish between these two teas and which one is better in terms of the content of particular compounds.
Author Response
Dear reviewer:
Thank you very much for constructive suggestions and comments, which greatly improve our manuscript entitled “A Comprehensive Investigation of Macro-Composition and Volatile Compounds in Spring-Picked and Autumn-Picked White Tea” (Manuscript ID: foods-1987872). All of the changes to the manuscript are indicated in the text using track changes. We wish that our answers could be as complete as possible to the question of reviewer.
Reviewer:
In the first paragraph, the authors reported the current state of knowledge on the differences between spring and autumn tea, pointing out that scientists have already proven that the timing of tea harvesting affects the qualitative and quantitative profile of the compounds it contains. The authors then moved on to introduce the news of white tea, which originated in China.
The last paragraph I consider the most important crowning the introduction of the article. The authors describe the purpose and scope of the paper while pointing out the procedures performed to compare the data collected in three consecutive years.
Details in manuscript are as follow:
line 106 - I propose to elaborate one again here the abbreviations SPWT and APWT, because the authors introduce the research material and the reader should know what it is without reading the introduction.
Author’s response: Thanks for your valuable comment. We have re-written in this part.
Details in manuscript are as follow:
Spring-picked was referred to as SP, and autumn-picked were called as AP. White teas picked in spring were called as SPWTs and those picked in autumn were called as AP-WTs. White peony teas and Shou Mei picked in spring were termed CMD19-21 and CSM19-21 respectively, while those picked in autumn were named QMD19-21 and QSM19-21. The number following CMD, QMD, CSM, and QSM represented the production years, corresponding to 2019, 2020, and 2021 respectively. The number after the year indicates a biological duplication (Table 1).
Table 1. White tea samples information
line 114-117 - And what was done with the buds and leaves after the brewing was finished? were discarded or did they remain in the brew? this needs to be specified in the text
Author’s response: Thanks for your valuable comment. We have re-written in this part.
Details in manuscript are as follow:
Tea infusions were prepared according to national standards (GB/T 23776-2018): each tea sample (3 g) was brewed with 150 mL boiling water for 5 min and discarded, after which tea infusions (30 mL) labelled with a three-digit code was presented to each panelist in a randomised order (Figure 1).
2.4. Macro-composition quantification – lines 124-133 - the authors must at least briefly describe the methodologies for analyzing each group of compounds. Chinese standards are not widely available in a language other than Chinese therefore this chapter does not allow other researchers to reproduce the methodology. Also, citing the methodology from another article is insufficient and the methodology should be at least briefly described.
Author’s response: Thanks for your valuable comment. We have re-written in this part.
Details in manuscript are as follow:
(1) Polyphenols were quantified using the Colorimetric method with Folin- Ciocalteu reagent based on the National Standards of China (GB/T 8312-2018). Specifically, 0.2 g of tea was added to 4.5 mL of 70% methanol (preheated to 70 ℃), after stirring and cool, centrifuge at 3500 r/min for 10 min, transfer the supernatant to a 10 mL test tube. Take 1mL of the supernatant into a volumetric flask and add 5 mL of Foline-Ciocalteau reagent and 4 mL of 7.5% Na2CO3, and was shaken for 8 min and made volume with pure water. The absorbance of the mixture was measured at 765 nm.
(2) Caffeine was quantified using the UV spectrophotometric method based on the National Standard of China (GB/T 8312-2013). Firstly, a 2 mL tea infusion was mixed with 4 mL of 0.01 mol/L hydrochloric acid and 1 mL basic lead acetate solution in a 100 mL volumetric flask. After added up to volume with water and mixed, the solution was filtered. The 12.5 mL filtrate was taken into 25 mL volumetric flask and added with 0.05 mL of 4.5 mol/L sulfuric acid to eliminate residual lead ion in the solution. After added up to volume with water and shaken, the mixed solution was filtered. Absorbance of the second filtrate was measured spectrophotometrically at 274 nm.
(3) The total free amino acids were quantified using the colourimetric method based on the National Standard of China (GB/T 8314-2013), with minor modifications. A 1 mL tea infusion and 0.5mL phosphate buffered saline were put into a 25mL test tubes, then 0.5 mL ninhydrin waw also added. The tubes were placed in a boiling water bath for 10 min, and then immediately cooled to room temperature in an ice bath. Finally, the absorbances of the samples was measured spectrophotometrically at 570 nm.
(4)TFs、TRs、TBs
The total content of theabrownins (TBs), theaflavins (TFs), and thearubigins(TRs) were determined using a previously discussed method[41,42]. Briefly, a total of 3g of dried tea was extracted with 125 mL boiling distilled water in a boiling water bath for 10 min and filtered, followed by cooled to room temperature. 50 mL of tea filtrate was pipetted in 50 mL of ethyl acetate with shaken for 5 min and the layers were separated after equilibration. A 0.8 mL portions of the ethyl acetate layer was made volume to 10 mL with 95% ethanol (solution A). Another 15 mL of the ethyl acetate layer was shaken with 15 mL of NaHCO3 solution (2.5%) for 30 s, stand until the layers separated. Pipetted 1.6 mL of ethyl acetate layer and fix the volume to 10 mL with 95% ethanol (solution C). A 0.8 mL sample of portions of the aqueous layer was diluted to 10 mL with 2.4 mL distilled water and 0.8 mL saturated oxalic acid solution (10.2%, m/v), and 6 mL of 95% ethanol (solution D). A 25 mL of tea filtrate was pipetted and mixed with 25 mL of butyl alcohol. After shaking for 3 min and the layers were separated after equilibration. A 0.4 mL sample of the aqueous layer (second) was made volume to 10 ml with 0.4 mL of the saturated oxalic acid solution, 1.2 mL of distilled water, and 95% ethanol (solution B). The absorbance of solutions A, B, C, and D at 380nm was measured spectrophotometrically using 95% ethanol as a blank. The results were calculated using the following formula:
TFs =100%
TRs = 100%
TBs= 100%
2.5. Identification and analysis of volatiles - how the authors identified the various volatile compounds? please write in the methodology
Author’s response: Thanks for your valuable comment. We have re-written in this part.
Details in manuscript are as follow:
Briefly, one quantitative ion and two to three qualitative ions were selected for each compound. All ions to be detected in each group were detected separately in the order of peak appearance, and if the retention times of the detected peaks were consistent with the standard reference, and if all the selected ions appeared in the mass spectra of the samples after subtraction of background, the substance was determined to be.
Reference
Yuan, H.; Cao, G.; Hou, X.; Huang, M.; Du, P.; Tan, T.; Zhang, Y.; Zhou, H.; Liu, X.; Liu, L.; et al. Development of a widely targeted volatilomics method for profiling volatilomes in plants. Mol. Plant 2022, 15, 189-202, doi: 10.1016/j.molp.2021.09.003.
3.2. Analysis of macro-composition of SPWTs and APWTs - the authors described the results of the work very poorly - mostly in the form of 1-3 sentences for each analysis. Most of the text is definitions and description of knowledge from the literature. There is also no explanation as to why any group of compounds was more or less in the infusions, and there is no discussion of the results with other studies.
Author’s response: Thanks for your valuable comment. We have re-written in this part.
Details in manuscript are as follow:
(1) Free amino acids are the basic units of protein and are organic compounds containing amine (-NH2) and carboxyl (-COOH) functional groups. They are one of the most important energy sources for plants and are the precursors for the biosynthesis of many important secondary metabolites. In this study, the free amino acid content of SPWTs was higher than that of APWTs, and this significant difference was observed in the 3-year samples (Figure 3B). Simultaneously, in the sensory evaluation results, the SPWTs scored higher than the APWTs in umami (Figure 2A, B). Free amino acids endow tea infusions with umami and smooth tastes, are the main contributors to the umami taste, and participate in the formation of aroma. This may cause the umami taste and smoothness of SPWTs to be stronger than those of APWTs. Liu et al. found that the nitrogen content stored in spring shoots is higher than in autumn, providing an abundant source of nitro-gen for amino acid synthesis. This may account for the difference in free amino acid content between SPWTs and APWTs. Additionally, with an increase in storage time, the content of free amino acids gradually decreased and was the highest in the samples with the shortest storage years, which was also reflected in the sensory evaluation results. These results corresponded well with previous findings among teas from different seasons.
(2) Alkaloids are small molecule nitrogenous compounds that, in tea trees, mainly include purine bases, pyrimidine bases, and pyridine bases, among which purine bases are the main components of alkaloids. Purine alkaloids have the same purine ring structure in tea trees, whereas purine bases are dominated by caffeine. Caffeine stimulates the central nervous system and enhances mental and physical processes in the human body. It accounts for 3% of the dry weight of tea leaves and is one of the most important factors in determining the quality of green tea, with a bitter taste that enhances the bitterness and astringency of the tea infusion. The caffeine content of SPWTs was higher than that of APWTs (Figure 3C). This may be due to the higher nitrogen content stored in spring shoots than in autumn, which promotes photosynthetic carbon assimilation, consumes more photosynthetic products, then facilitates the synthesis and accumulation of caffeine. This may result in a stronger astringency and thickness in SPWTs than that in APWTs. The caffeine content does not show a significant trend of increase or decrease with increasing storage time, which was due to the stable structure of the purine ring of caffeine.
(3) Polyphenols are compounds with one phenolic ring (phenolic acid/phenolic alcohol) or multiple aromatic rings with one or more hydroxyl groups and have been widely studied. In our research, the total content of polyphenols in the white peony and Shou Mei tea samples from the same year was higher in SP than in AP samples (Figure 3D), and with the increase in time, the content showed a downward trend, which is similar to the findings of previous studies. The total polyphenol content of the same white tea was significantly different between SPWTs and APWTs of the same year. Tea polyphenols are associated with bitterness in sensory evaluation and enhancement of the astringency of tea infusions, which may be one reason why SPWTs have stronger astringency and thickness than APWTs. The polyphenol content of tea tends to decrease as storage time increases. Technically, the decrease in tea polyphenols content with storage age may be due to the Folin-Ciocalteu reagent not reacting with the phenolic hydroxyl groups of polymeric polyphenols (e. g. TFs, TRs, and TBs).
(4) Flavonoids are a large class of structurally diverse analytes composed of a variety of basic skeletons and a series of derivatives. They are also the most representative secondary metabolites in tea and include flavonoids, flavonols, flavanones, flavanols, and anthocyanins. In this study, the flavonoid content of APWTs was higher than that of SPWTs (Figure 3E), and the difference between SP and AP samples was significant. Zhang et al. found the content of flavonoids increased at a higher temperature under the same growth conditions. Additionally, by checking historical temperatures, the average temperature in autumn is higher than that in spring, which causes the content of flavonoids to be higher in autumn than that in spring (Supplementary Table S2). Tea flavonoids have attracted extensive attention because of their multiple roles in improving the resistance of fresh tea leaves to multiple stresses and forming unique flavours and colours in tea infusions, which enhance the bitterness of tea infusions. This may result in a darker colour in APWTs than in SPWTs (Figure 1).
(5) TFs, TRs, and TBs are a range of catechin derivatives, which are also the main water-soluble pigments in tea infusions and affect their taste. TFs are compounds with a benzotropinone structure that are golden yellow in colour, and pungent and astringent in flavour. They are an important component of the strength of taste and crispness of the tea infusion and are related to astringency. TRs are complex bronzing phenols with a sweet, mellow flavour, which is an important component for the consistency and strength of the tea broth. TBs are dark brown macromolecular compounds, mainly produced by the oxidation of TRs and TFs, and are negatively correlated with black tea infusions’ colour and taste. Here, we found that the TF (Figure 3G) and TR (Figure 3H) contents were higher in SPWTs than in APWTs in the first two years, but the difference was not significant in the third year. The TB (Figure 3I) content was lower in SPWTs than in APWTs in the latter two years, whereas the difference was not significant in the first year. The combined effects of TFs, TRs, and TBs may contribute to differences in the production of SPWTs and APWTs. The content of TBs showed a general increasing trend co with the de-crease in tea polyphenols, suggesting that increased storage time enhances the polymerisation of tea polyphenols and leads to an increase in the content of TBs with storage. This result was in accordance with the findings by Zhao, et al.
Figure 3 - The authors did not specify which year of the 2019-2021 period these results come from.
Author’s response: Thanks for your valuable comment. We have re-written in this part. Explanation of the sample code we have added in the figure notes.
Details in manuscript are as follow:
White teas picked in spring were called as SPWTs and those picked in autumn were called as APWTs. White peony teas and Shou Mei picked in spring were termed CMD19-21 and CSM19-21 respectively, while those picked in autumn were named QMD19-21 and QSM19-21. The number following CMD, QMD, CSM, and QSM represented the production years, corresponding to 2019, 2020, and 2021 respectively. The number after the year indicates a biological duplication (Table 1).
Figure 4 A,B,C - The images/diagrams are unreadable, nothing can be seen on them.
Author’s response: Thanks for your valuable comment. We have increased the resolution
Details in manuscript are as follow:
Figure 5 - I recommend removing the heat map figure because it is very unreadable and adds little information to the article.
Author’s response: Thanks for your valuable comment. We have removed Figure 5.
3.4.1. PCA of VOCs in SPWTs and APWTs - The authors did not describe in detail what the chart shows, what the classification looks like. Are there any regularities about the timing or years of harvesting?
Thanks for your valuable comment. We have re-written in this part.
Details in manuscript are as follow:
The PCA based on all the identified VOCs was performed to analyze the confidence of the identification results and the overall compound differences between the 12 groups (Figure 5A). The first two principal components (PCs) accounted for 55.09% of the total variation (PC1=30.19%, PC2=24.9%). The SPWTs were distributed on the positive side of the first component and the APWTs were distributed on the negative side of the first com-ponent. CSM19 and QSM19 were relatively independent of each other in the PC2 direction. Observing the loading plot (Figure 5B), a total of 189 VOCs were in the third quadrant, and the markers a, b, c, d, and e contributed greatly to the differences between groups, namely (1R,4S,5S)-1,8-dimethyl-4-prop-1-en-2-yl-spiro[4.5]dec-8-ene, p-Tolyl isobutyrate, 1-(Furan-2-yl)-2-methylpentan-1-one, 4-(1-methylethyl)-Benzaldehyde, and 2,3,6-trimethyl phenol. P-Tolyl isobutyrate has a “green” and “floral” aroma; 4-(1-methylethyl)-Benzaldehyde has a “green” and “herbal” aroma. The difference in their content between sample groups may be responsible for the stronger “fresh”, and “flower” aroma of CSM19 than QSM19. Overall, the samples were well differentiated based on the classification of seasons and were clustered using three biological replicates, indicating that the volatile content in different seasons varied greatly.
line 356- Merad reaction? what is this?
Author’s response: We are sorry for our careless mistake. We have re-written in this part. Details in manuscript are as follow:
Maillard reaction
Figure 7 - Like the other charts it is poorly readable and if there were no title it would be impossible to know what it represents.
Author’s response: We are sorry for our careless mistake. We have increased the resolution.
Details in manuscript are as follow:
line 418-420- this is a repetition of information already given earlier therefore I recommend removing this text.
Author’s response: Thanks for your valuable comment. We have removed this duplicate information
The authors made a great deal of determinations but gave very few conclusions. There is no reference to the seasonality of the harvest and its impact on the formation of non-volatile compounds. In addition, they did not state how spring and autumn teas differ (volatile and non-volatile compounds). They did not state what needs to be followed to distinguish between these two teas and which one is better in terms of the content of particular compounds.
Author’s response: Thanks for your valuable comment. We have re-written in this part.
Details in manuscript are as follow:
In this study, sensory evaluation, biochemical composition determination, GC-MS, OAV analysis, and multivariate statistical analysis were combined to measure the macro-compositions and VOCs of SPWTs and APWTs. The “umami”, “smoothness”, “thickness”, “sweetness” taste, “flower” and “fresh” aromas were stronger in SPWTs than that in APWTs. The content of free amino acids, caffeine and tea polyphenols in SPWTs were higher than that in APWTs, while the flavonoids content is lower than APWTs. The results of OPLS-DA showed that free amino acids, tea polyphenols, caffeine, flavonoids, and TFs were important macro-components for distinguishing SPWTs from APWTs. The difference in their content leads to a difference in taste between SPWTs and APWTs. A total of 545 VOCs were detected by GC-MS, and the key compounds were screened by stoichiometry, OAV analysis, and OPLS-DA analysis. The results indicated that (2E,4E)-2,4-octadienal and (-)-cis-rose oxide were the key VOCs that distinguished SPWTs from APWTs. The aroma of (2E,4E)-2,4-octadienal is described as "green" and the aroma of (-)-cis-rose oxide is described as "rose”, “green” and “floral". Their higher content in SPWTs than APWTs is responsible for the stronger “flower” and “fresh” aroma of SPWTs than APWTs. In the future, the molecular regulatory mechanisms underlying the quality differences between SPWT and APWT will be further explored. This study contributes to an in-depth understanding of the taste and aroma of white tea, and provides a scientific basis for future work to improve the quality of white tea.

Round 2
Reviewer 1 Report
Dear authors,
thank you so much for submitting the revized version of the manuscript entitled "A Comprehensive Investigation of Macro-Composition and Volatile Compounds in Spring-Picked and Autumn-Picked White Tea".
I am generally satisfied with the modifications. I would like to point out some elements that require corrections.
Line 50 : Please replace the word "tea" with the intended word.
Line 99: What is a "review" glass ? I was unable to access the reference cited. I suggest to remove this element that I juge as unclear.
Line 189: Please remove the capital letter of the word "colorimetric".
Line 191 : for any occurence of percentage in the manuscript, please precise when adequate if it refers to %(m/v) or %(v/v).
Line 192: Please replace "r/min" by "rpm". Or, it would be even better to replace by the value in g if possible.
Line 200: Please favor the followign formatting "mol.L-1" in the whole manuscript.
Line 218: Please define "DeltaA" and "W".
Line 221: Please put a capital letter to "The".
Lines 239-241: Please define EA, EB, EC and ED.
Supplementary Figure S1 : sweet aroma is present twice in Supplementary Figure S1. One of the parameters shows systematic higher intensity in SPWT than APWT, and this should be added to the commentary. Also, please change the title to "results of the ANOVA of the sensory taste and aroma attribute scores" and explain the meaning of the letters (in the title).
Lines 383-384: For clarity, you can add an additional sentence to explain that polymerization phenomenon takes please during storage.
Figure 5A: The color inside the ellips make it difficult to recognize the color of the dots, please remove the color inside and favor coloring the outer lines instead.
Line 505: I do not understand the meaning of "relatively independent of each other in the PC2 direction".
Line 507: please explain why the a,b,c and d parameters contributed more than others ? Display the necessary data (as supplementary material for example) if needed.
Line 545: Do you mean "dominated in number" ?
Line 557: Please change to "polymerization".
Reference: Please provide all DOIs when available.
Therefore, I request minor revisions of the manuscript. If treated appropriately, the editor may accept the newly revized manuscript for publication.
Reviewer 2 Report
The authors have addressed all comments and feedback. They made all the necessary corrections to the text, enlarged the discussion of the results and re-stressed the conclusions. Also visually, the presentation of the results has been significantly improved.
I have no further comments on the content of the manuscript.